# Detecting directed motion and confinement in single-particle trajectories using hidden variables

**François Simon[1]\*, Guillaume Ramadier[1], Inès Fonquernie[1,2], Janka Zsok[3], Sergiy Patskovsky[1], Michel Meunier[1], Caroline Boudoux[1], Elisa Dultz[3], Lucien E Weiss[1]\***

[1]Department of Engineering Physics, Polytechnique Montréal, Montréal, Canada; [2]École Polytechnique, Palaiseau, France; [3]Institute of Biochemistry, ETH Zürich, Zürich, Switzerland

## eLife Assessment

In this **valuable** contribution, the authors present a novel and versatile probabilistic tool for classifying tracking behaviors and understanding parameters for different types of single-particle motion. The software package will be broadly applicable to single-particle tracking studies. The methodology has been **convincingly** tested by computational comparisons and experimental data, although the mathematical foundation for the hypothesis testing method can be further strengthened.

**Abstract** Single-particle tracking is a powerful tool for understanding protein dynamics and characterizing microenvironments. As the motion of unconstrained nanoscale particles is governed by Brownian diffusion, deviations from this behavior are biophysically insightful. However, the stochastic nature of particle movement and the presence of localization error pose a challenge for the robust classification of non-Brownian motion. Here, we present *aTrack*, a versatile tool for classifying track behaviors and extracting key parameters for particles undergoing Brownian, confined, or directed motion. Our tool quickly and accurately estimates motion parameters from individual tracks. Further, our tool can analyze populations of tracks and determine the most likely number of motion states. We show the working range of our approach on simulated tracks and demonstrate its application for characterizing particle motion in *Saccharomyces cerevisiae* and for biosensing applications in *Escherichia coli*. aTrack is implemented as a stand-alone software, making it simple to analyze track data.

## Introduction

Single-particle tracking (SPT) is a valuable tool for characterizing protein activity (***Shen et al., 2017***; ***Kapanidis et al., 2018***; ***Wang et al., 2021***). While Brownian diffusion is commonly observed inside cells, deviations from this behavior are of major interest and can provide key biophysical insight (***Saxton, 2007***; ***Höfling and Franosch, 2013***; ***Metzler et al., 2014***). Classically, non-Brownian behaviors have been identified by fitting a power law to the mean-squared displacements (MSD) as a function of the time lag $\tau$ ($MSD = \Gamma \cdot \tau^{\alpha}$; ***Metzler et al., 2014***), where sub-diffusive motion has an exponent $\alpha < 1$ and super-diffusion has an $\alpha > 1$. While fitting the MSD curve can therefore be used for categorizing motion, the anomalous exponent and the generalized diffusion coefficient provide little insight into the combination of underlying biological forces, for example diffusion, directed motion, and confinement.

**\*For correspondence:** simon.francois@protonmail.com (FS); lucien.weiss@polymtl.ca (LEW)

**Competing interest:** The authors declare that no competing interests exist.

**Preprint posted** 21 April 2024
**Sent for Review** 11 May 2024
**Reviewed preprint posted** 30 July 2024
**Reviewed preprint revised** 04 September 2025
**Version of Record published** 07 April 2026

To accurately model the motion encountered in cellular contexts, physics-based models have been developed to explicitly describe interactions that give rise to nonlinear MSDs. For example, super-diffusion can arise from the linear movement powered by molecular motors (*Monnier et al., 2015*), and sub-diffusive motion can arise from confinement (*Bernstein and Fricks, 2016*; *Slator and Burroughs, 2018*). In such models, the observed track positions result from a stochastic process described by hidden (unobserved) variables that evolve with time.

In the context of random processes, Maximum Likelihood Estimation (MLE) is used to determine the underlying parameters of the model. MLE consists of computing the probability of observing a track given the model and its parameters. The main challenge in models with hidden variables is that they require computing the integral of a joint probability density over all possible hidden variables. The few tools that use this type of models perform this integration step using either coarse-grained approximations (*Monnier et al., 2015*) or sampling methods that are quite slow and inaccurate (*Slator and Burroughs, 2018*; *Bernstein and Fricks, 2016*). Thus, designing a hidden-variable method that benefits from accurate and efficient integration is an important challenge for improving the reliability and ease of use of the physics-based models. Further, when the underlying physical model is undecided, statistical tests can be applied to identify the most appropriate model (*Burnecki et al., 2012*; *Briane et al., 2018*; *Woringer et al., 2020*). The current hidden-variable models applied to non-Brownian diffusion have two additional disadvantages: They are specialized in either confinement or directed motion but do not treat both types of motion, and they typically do not allow variations of the potential well position or the speed of the directed motion, properties that are frequently encountered in biological systems.

Here, we present aTrack, an analysis tool that alleviates the limitations mentioned above. Our approach uses a versatile motion model that considers the relationships between the observed track, the real particle positions (localization error and Brownian motion), as well as the influence of a non-Brownian variable that can either be the potential well for confined diffusion or the velocity vector for directed motion. The main innovations of this model are: (1) it uses analytical recurrence formulas to perform the integration step for complex motion, improving speed and accuracy; (2) it handles both confined and directed motion; (3) anomalous parameters, such as the center of the potential well and the velocity vector, are allowed to change through time to better represent tracks with changing directed motion or confinement area; and lastly, (4) for a given track or set of tracks, aTrack can determine whether tracks can be statistically categorized as confined or directed, and the parameters that best describe their behavior, for example, diffusion coefficient, radius of confinement, and speed of directed motion.

We validate the approach on simulated data and demonstrate its versatility for analyzing a variety of experimental SPT data, including particle diffusion in an optical trap, detection of motile bacteria with gold nanoparticles, and motion characterization of spindle pole bodies in budding yeast.

## Results
### Accounting for hidden variables that characterize confined and persistent motion
#### Modeling non-Brownian motion
We model noisy tracks undergoing confined, Brownian, and directed motion by considering four relations at each time step: (1) there is a Brownian diffusion step followed by (2) an anomalous step (*Figure 1a–b*); (3) the hidden anomalous variable, $h$, can evolve according to a Gaussian distribution; and (4) localization error is incorporated as a Gaussian-distributed noise term added to the underlying real position to produce the observed positions. This model encompasses a variety of motion types depending on the model parameters, as illustrated in *Figure 1c*. For example, particles can be immobile, where the observed displacements are only due to localization error; tracks can undergo Brownian motion, as well as anomalous super- and subdiffusion; or multiple motion mechanisms can also occur simultaneously (e.g. diffusion and drift, or changes in directed motion direction and speed). The model can also account for confinement in a quadratic potential well (*Bernstein and Fricks, 2016*; *Oswald et al., 2016*; *Slator and Burroughs, 2018*), which is either fixed or diffusing. In our model, velocity is a characteristic parameter of directed motion and the confinement factor represents the force within a potential well. More precisely, the confinement factor $l$ is defined such that at each

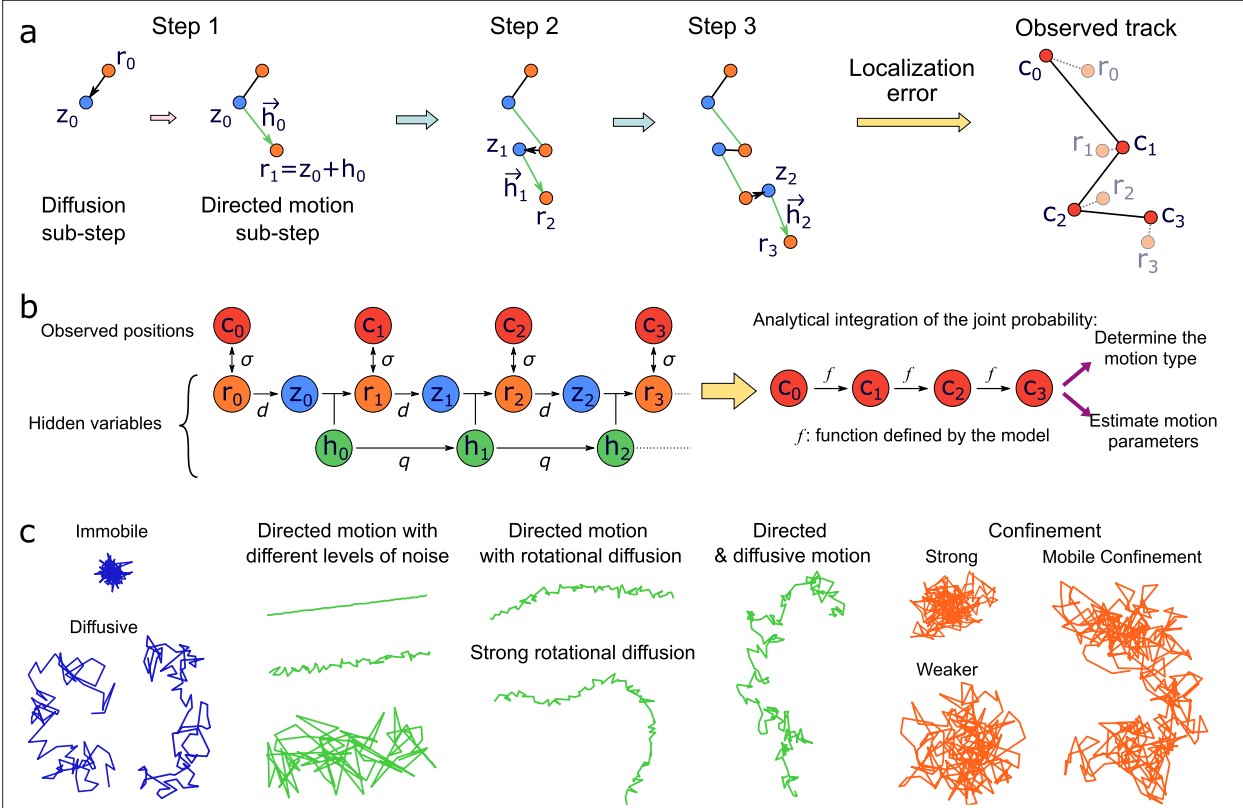

**Figure 1.** Principle of *aTrack*. (**a**) The track-generation steps with our motion model shown here for directed motion. Each time step is decomposed into sub-steps, namely diffusion and anomalous motion (either directed or confining) with $r_i$ the real positions, $z_i$ an intermediate position, $h_i$ the anomalous variable, and the generation of observed (measured) positions, $c_i$. The variables $c_i - r_i$ and $r_{i+1} - r_i$ follow Gaussian distributions with mean 0 and standard deviations $\sigma$ and $d$, respectively. The sub-step from $z_i$ to $r_{i+1}$ is deterministic and the anomalous variable $h_i$ can also evolve with a standard deviation $q$. (**b**) Graph representation of aTrack showing the motion model (*left*), analytical integration (*middle*), and outputs (*right*). To compute the track probability, we integrate over the hidden variables. This results in an analytical recurrence formula that is used to determine the type of motion and to estimate the parameters of the motion. (**c**) Examples of tracks that can be produced with our motion model.

time step the particle position is updated by $l$ times the distance particle/potential well center (see the Methods section for more details).

## Determining the type of motion

To categorize the type of motion from a measured trajectory, we first calculate the likelihood that a track belongs to each considered motion class (diffusion, directed, or confinement) and then perform a statistical comparison between the likelihoods. To do the former, we must integrate the joint probability density function of a track for all the modeled hidden variables, for example, all the real positions and the potential well positions or the velocity vectors. Since our model is a multivariate Gaussian process expressed as the product of univariate Gaussian functions, we can perform the integration step using analytical recurrence formulas (see Supplementary information; *Relich et al., 2016*; *Simon et al., 2023*; *Simon and Cardona, 2025*). The recurrence formula enables our model computation time to scale linearly with the number of time points.

We can apply these formulas to determine the best sets of parameters and the maximum likelihoods, then use the ratio between the maximum likelihood assuming Brownian diffusion (null hypothesis) and the maximum likelihood assuming confinement or directed motion as the alternative hypothesis to build a likelihood ratio test (*Wilks, 1938*; *Van der Vaart, 2000*). *Figure 2—figure supplement 2* shows that these likelihood ratios, $\rho = l_{\text{Brownian}}/l_{\text{confined}}$ or $\rho = l_{\text{Brownian}}/l_{\text{directed}}$ are systematically skewed towards 1 when particles follow Brownian diffusion. Conversely, $\rho$ is skewed toward zero when applying the directed-diffusion test to directed tracks or the confined-diffusion test to confined tracks. These properties enable the likelihood ratio to serve as a robust proxy for the p-value as it

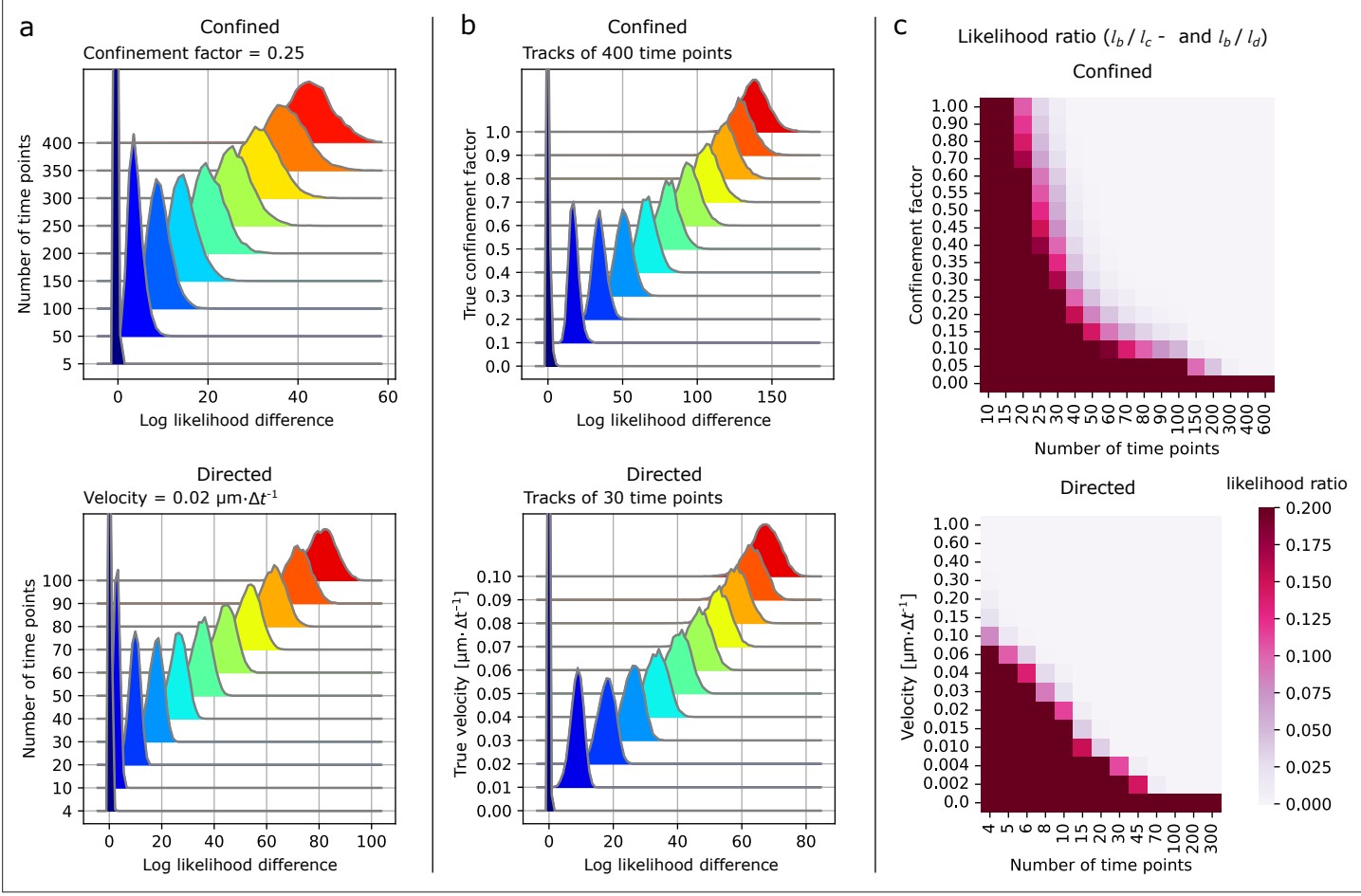

**Figure 2.** Determining the motion type using a likelihood-ratio test. (**a, b**) Probability distributions of the difference between the log of the maximum likelihood of the alternative hypothesis (either confinement $L_c$ or directed $L_d$) and the null hypothesis (Brownian diffusion $L_b$) for single tracks (10,000 tracks). Confinement factor $l = 0.25$ and velocity $v = 0.02$ µm·$\Delta t^{-1}$. (**a**) Effect of the number of time points in a track on its log difference ($L_c - L_b$ for confined tracks) and ($L_d - L_b$ for directed tracks). (**b**) The ability to distinguish confinement and directed motion from diffusion as a function of the confinement factor and particle velocity, respectively. (**c**) Heatmaps of the likelihood ratios $l_b/l_c$ (confined) or $l_b/l_d$ (directed) varying both the anomalous diffusion parameter and the track length. Mean of 10,000 tracks. (**a-c**) When not stated otherwise, the track parameters were as follows. Localization error σ = 0.02 µm . Confined tracks: diffusion length $d = 0.1$ µm. Directed tracks: $d = 0.0$ µm (no diffusion), constant speed and orientation.

The online version of this article includes the following figure supplement(s) for figure 2:

**Figure supplement 1.** Log-likelihood differences as a function of the track length.

**Figure supplement 2.** Likelihood ratios as a function of the track length.

overestimates it. As a consequence, if $\rho < X$ with $X$ the type I error rate (e.g. 0.05), we can reject the null hypothesis with a confidence of $1 - X$. Relying on the skewness of the likelihood ratio to obtain an upper bound of the p-value is a simple way to categorize the type of motion, but sometimes a more sensitive test is needed. In such cases, one can use simulations to better estimate the p-value (See Supplementary information: Statistical test for more details).

To estimate the impact of the track length on the classification, we simulated tracks of varying lengths (5–400 steps for confined, 4–100 steps for directed) undergoing either confined diffusion or linear motion (without diffusion) and computed the likelihood ratios. As expected, the classification certainty increases with the track length, where the inverse logarithm of $\rho$ increases with the number of steps (*Figure 2a*). Of course, the statistical certainty depends on the track parameters (*Figure 2b*). In the range of evaluated anomalous parameters, the significance of the test increased with the confinement factor and the directed motion velocity. To determine the useful range more systematically, we varied the track length and either the confinement factor for confined motion or the velocity for directed motion and computed the average likelihood ratios (*Figure 2c*). A low average ratio indicates

significantly low p-values for most tracks, as this ratio is an overestimate of the p-value. We see that the test is significant as long as the anomalous parameter is high enough or the track length is high enough. Note that increasing the confinement factor so much that the confinement radius becomes similar to or smaller than the localization error will impair the capacity of the test.

## Characterizing confinement

To characterize confined trajectories, aTrack estimates several parameters, namely the diffusion coefficient $D$ and the diffusion length $\sqrt{2D\Delta t}$ where $\Delta t$ is the single-frame time step; the confinement factor

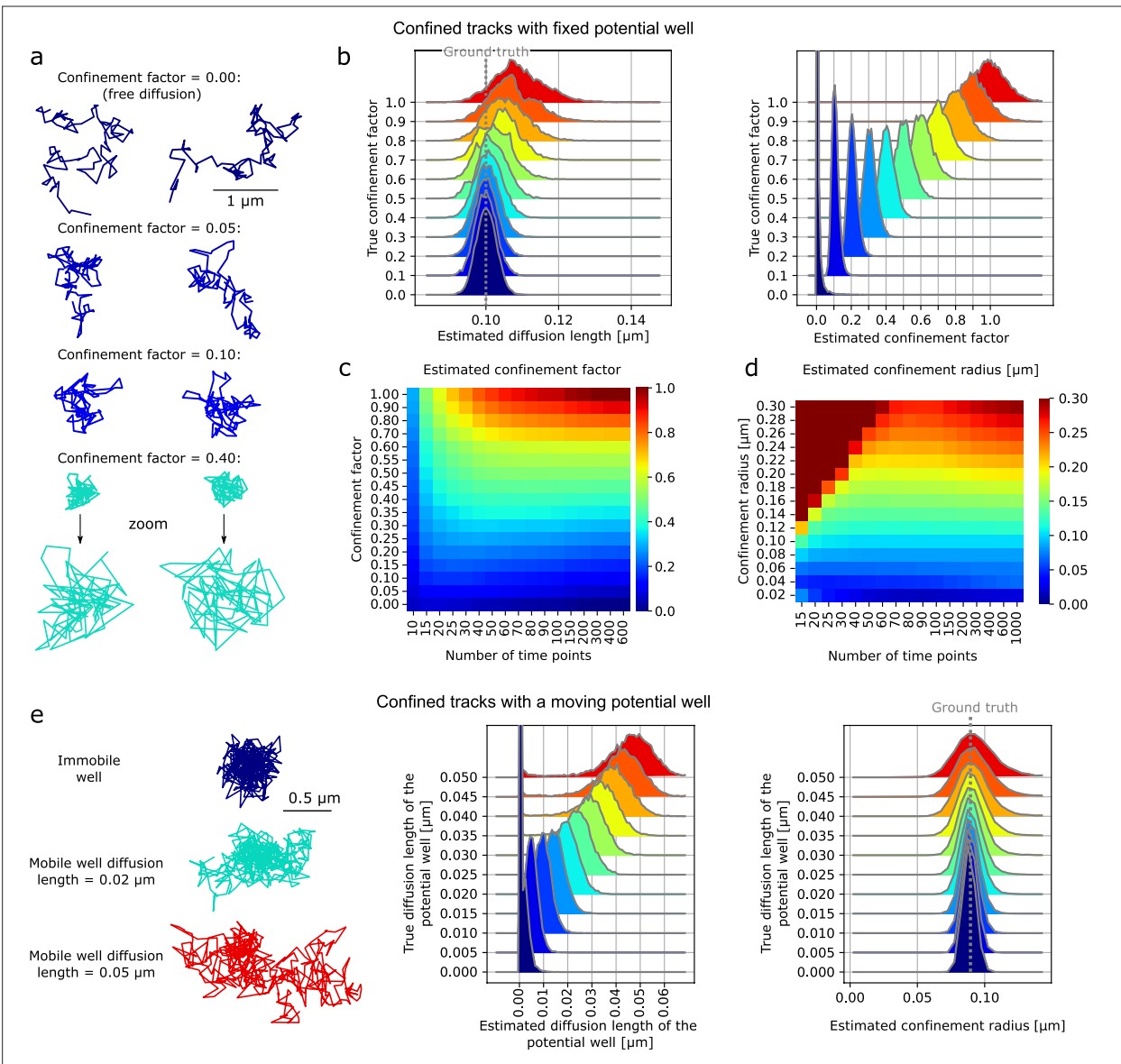

**Figure 3.** Characterizing confinement with *aTrack*. (**a-d**) Confinement of tracks with a fixed potential well. (**a**) Examples of simulated tracks with different confinement factors. (**b**) Histograms of the estimated parameters for individual tracks of 200 time points varying the number of time points in tracks. (**c-d**) Heatmaps of the mean estimated confinement factor and confinement radius depending on the track length and the confinement factor (per time step) or radius, respectively. (**e**) Confinement of tracks with a moving potential well (Brownian motion). Left: simulated tracks with different diffusion lengths of the potential well. Right: histograms of the estimated diffusion length of the potential well and confinement radius = $\sqrt{D\Delta t/l}$ varying the actual diffusion length of the potential well. Confinement factor = 0.1 per time step. (**a-d**) 10,000 tracks per condition. $d$ = 0.1 μm, Localization error σ = 0.02 μm . See *Figure 3—figure supplement 1* for complementary results.

The online version of this article includes the following figure supplement(s) for figure 3:

**Figure supplement 1.** Parameter estimates for confined motion model.

$l$, which is proportional to the spring constant of the potential well; the diffusion coefficient of the confinement area $D_c$; and the localization error $\sigma$; and the confinement radius, which is proportional to $\sqrt{D\Delta t/l}$, can also be calculated (see the Methods section for more details).

To measure the precision of parameter estimation, we simulated tracks with different confinement factors, $l$, (*Figure 3a*). We then used aTrack to estimate the parameters for each track of 200 time steps (*Figure 3*). The estimates of the diffusion coefficient, confinement factor, and confinement radius were accurate over the range of confinement factors, 0–1 per time step. Panel *Figure 3c* shows the working range for calculating the confinement factor as a function of track length and confinement factor. Longer tracks result in better parameter estimation. Similarly, our method correctly estimates the confinement radius (*Figure 3—figure supplement 1d*). In a second set of simulations, we tested the reliability of our predictions when the confinement area is moving (*Figure 3d*, *Figure 3—figure supplement 1e*).

## Characterizing directed motion

Super-diffusive behavior, in particular directed motion, occurs in a variety of circumstances, such as molecular motor-mediated active transport (*Pierobon et al., 2009*; *Monnier et al., 2015*) and polymerase processivity (*van Teeffelen et al., 2011*). Localization error complicates velocity estimation, especially when the localization error is relatively large. We simulated noisy tracks undergoing linear motion at various speeds (*Figure 4a*) with a localization error of 20 nm·$\Delta t^{-1}$ and estimated the velocity per track. *Figure 4b* shows the velocity estimates for tracks with 30 time points. By varying both the track length and the velocity, we found our method to be reliable for a wide range of velocities as long as tracks were long enough (*Figure 4c*, *Figure 4—figure supplement 1*).

In real experiments, persistent motion is rarely perfectly linear, that is, changes in direction and speed are very common (*Pilling et al., 2006*; *Pierobon et al., 2009*; *Monnier et al., 2015*; *van Teeffelen et al., 2011*). Naturally, characterizing directed motion with direction (orientation) changes is more difficult when localization error is non-negligible (*Calderon et al., 2014*; *Bouzigues and Dahan, 2007*). To verify our method's capacity to accurately quantify tracks with such behaviors, we simulated tracks with constant speed and random changes of orientation (rotational diffusion). We previously observed similar behavior when analyzing the directed motion of the Rod complex in bacteria (*Özbaykal et al., 2020*), motor-driven directed motion in mammalian cells (*Calderon et al., 2014*; *Tsitkov et al., 2020*), and cell motility (*Ping, 2012*). We varied the rate of orientation changes and the track velocity to determine the working range of aTrack for this type of directed motion (*Figure 4d, e and f*). In *Figure 4e and f*, we find accurate estimates of the velocity and the rotational diffusion for a wide range of parameters. As the velocity increases, we can distinguish directed motion from Brownian motion for an increasing range of rotational diffusion (*Figure 4e*). However, we also see that fast changes in orientation (high rotational diffusion) make estimating the velocity and rotational diffusion more difficult. This tradeoff is expected, as rapid changes in direction make the track appear more diffusive, as shown in the likelihood ratio heatmap *Figure 4—figure supplement 2a*, left panel. The diffusion length heatmap (*Figure 4—figure supplement 2a*, right panel) explains why the rotational diffusion coefficient is poorly estimated when high: the model interprets the high rotational diffusion as simple diffusion since the two types of motion are very difficult to distinguish in this regime.

Sometimes, particles undergo diffusion and directed motion simultaneously, for example, particles diffusing in a flowing medium (*Qian et al., 1991*). Sample drift can also introduce a combination of diffusion and directed motion in single-molecule tracking. Indeed, the thermal expansion of instrument components like microscope stages can induce steady motion that masks the biologically relevant diffusive motion (*Fazekas et al., 2021*). We first tested that our method could correctly differentiate directed motion from diffusion (*Figure 4g*, *Figure 4—figure supplement 2b*) for a range of diffusion coefficients and track lengths at a fixed velocity of 0.1 μm.$\Delta t^{-1}$. For mixed motion, our method accurately estimates the diffusion length and velocities even for short tracks, provided the diffusion length was low compared to the velocity. When the diffusion length is large compared to the velocity, parameters can still be predicted, but longer tracks are needed for reliable estimates.

## Characterizing populations of tracks

The amount of information in an individual track is limited by its length, making it difficult to extract the parameters precisely. One way to overcome this limitation is to consider a population of tracks

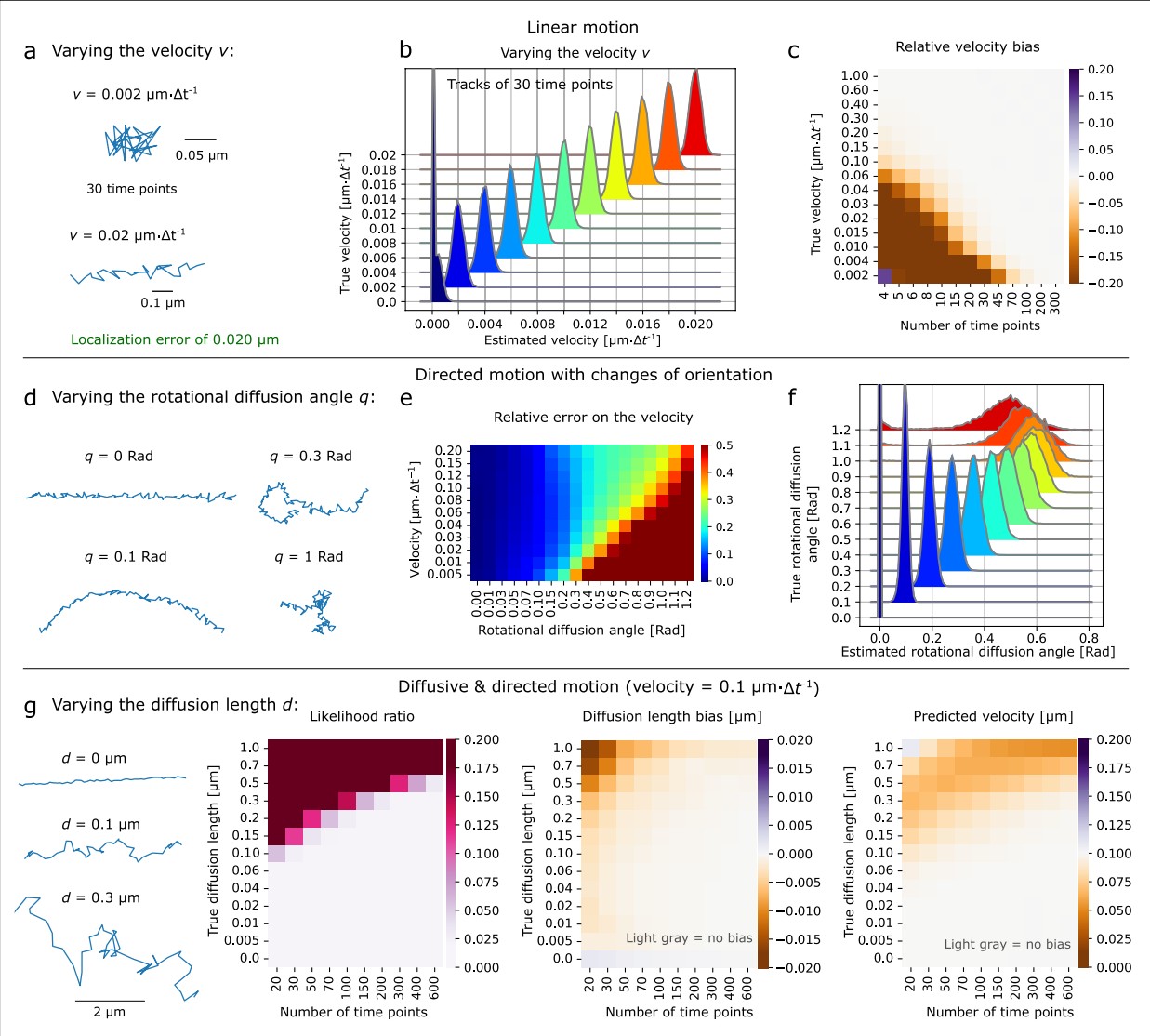

**Figure 4.** Characterizing directed motion with *aTrack*. (**a-c**) Tracks with linear motion (constant speed and orientation). a: Examples of simulated tracks in directed motion. (**b**) Histograms of the estimated velocity of individual tracks of 30 time points. 10,000 tracks per histogram. True parameters $d = 0.\mu m$, localization error $\sigma = 0.02$ µm (fixed). The next panels use the same parameters unless specified otherwise. (**c**) Heatmap of the relative biases on the estimated velocity ($\frac{v_{est} - v_{true}}{v_{true}}$). (**d-f**) Tracks with constant speed but changing orientation. d: Simulated directed tracks with rotational diffusion. Here, the rotational diffusion angle coefficient is defined as the standard deviation of the change of orientation at each time step (analogous to the diffusion length), $v = 0.02$ µm·$\Delta t^{-1}$. (**e**) Heatmap of the error on the rotational diffusion angle for a range of velocities and rotational diffusion angles. Tracks of 200 time points. (**f**) Distributions of the estimated rotational diffusion angle for a range of rotational diffusion angles. Tracks of 200 time points with $v = 0.1$ µm·$\Delta t^{-1}$. (**g**) Tracks simultaneously undergoing both linear motion and diffusion with varying levels of diffusion. Heatmaps of the likelihood ratio, bias on the diffusion length in µm, and estimated velocity depending on the number of time points per track and on the diffusion length $d$. (**a-g**) Where not stated otherwise, the track parameters were as follows. Localization error $\sigma = 0.02$ µm, $d = 0.0$ µm, velocity $v = 0.1$ µm $\Delta t^{-1}$, constant speed and orientation.

The online version of this article includes the following figure supplement(s) for figure 4:

**Figure supplement 1.** Linear motion model estimations.

**Figure supplement 2.** Impact of directional changes in directed motion models.

---

that have the same state. The likelihood of the population can be easily computed by multiplying the probabilities of the individual tracks. To test our population approach, we simulated two populations: confined tracks and directed tracks. Doing so, we demonstrated that the log likelihood increases linearly with the number of tracks and that using more tracks results in more precise parameter estimates (*Figure 5—figure supplement 1*).

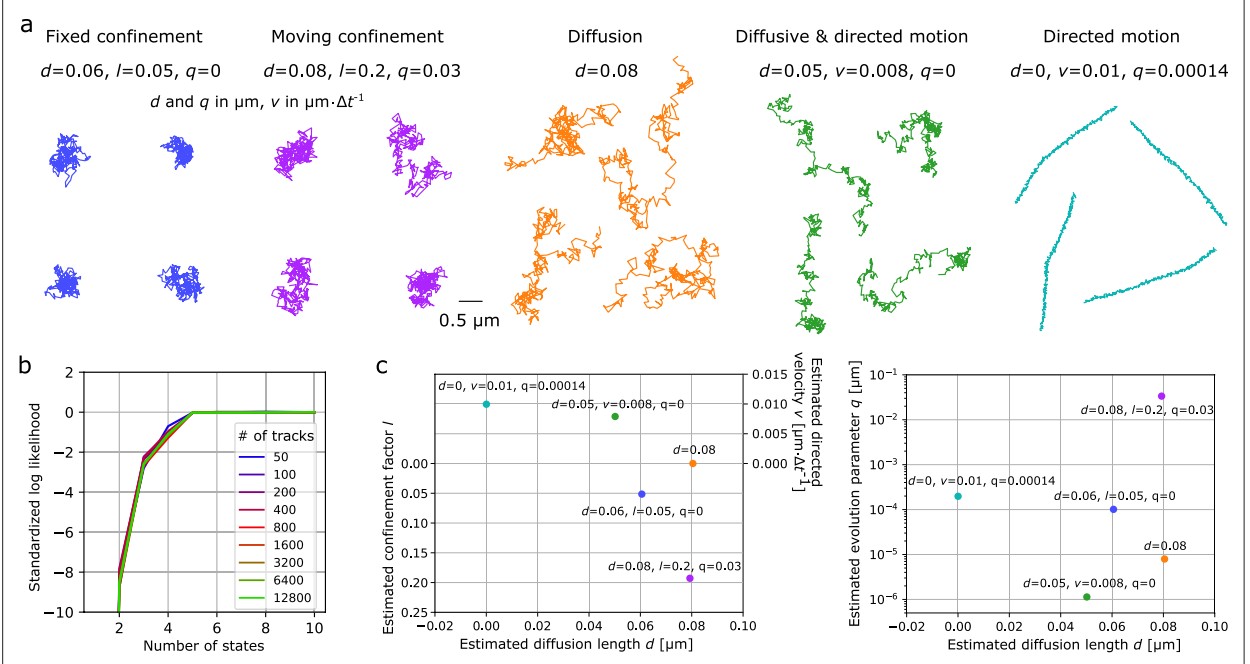

**Figure 5.** Characterizing populations of multiple states. Analysis of tracks with five sub-populations of set diffusion length $d$, confinement factor $l$, velocity $v$ for directed tracks, and anomalous change parameter $q$ (diffusion length of the potential well for confined tracks and changes of speed for directed tracks). Tracks are 300 time points long. (**a**) Track examples from each of the five states with the corresponding state parameters. (**b**) Log likelihood of the model depending on the number of states assumed by the model. The log likelihood was normalized by the number of tracks, offset by the log likelihood assuming 10 states. (**c**) Estimated parameters for the five states (using a five-state model).

The online version of this article includes the following figure supplement(s) for figure 5:

**Figure supplement 1.** Dataset size and parameter estimate error.

**Figure supplement 2.** Estimating the number of states.

Populations of tracks can often contain multiple states (e.g. free diffusion and directed motion). By taking into account the fraction of the particle in each state, we built a multi-state population model. We tested the capacity of our multistate population model on groups of simulated tracks with 300 time points, where each track follows one of the five states shown in *Figure 5a*.

The first step is to determine the number of states. To this end, we compute the likelihood of the model depending on the number of states. As expected, the likelihood increases with the number of states, until the correct number of states is reached and the function plateaus, in this case five states (*Figure 5b*). This plateau in likelihood is a good indicator that the appropriate number of states is reached; however, increasing the number of states further usually results in higher likelihood, albeit marginally.

Quantitative criteria such as the Akaike Information Criterion (AIC) *Akaike, 1974* and the Bayesian Information Criterion are often used to determine the number of parameters of a model by placing a trade-off between increasing the likelihood and increasing the number of parameters. When increasing the number of tracks from 50 to 12,800, we found these criteria to be unreliable (*Figure 5—figure supplement 2*). In theory, these criteria only work if the true underlying model is included in the alternative models, which is never the case for real tracks. As an alternative, we found that adding a small penalization term proportional to the number of parameters and the log likelihood provides a reliable criterion for identifying the number of states for any dataset size, even with mismatches between the data and the model assumptions.

Once we have identified the number of states, the parameters of each state are estimated at the population level. We found accurate parameter estimations for all the states, even when individual tracks remain difficult to classify (*Figure 5b*). For instance, classifying whether a track is in the diffusive state (orange) or diffusive plus directed state (green) is intrinsically hard due to the similarity in their motion patterns.

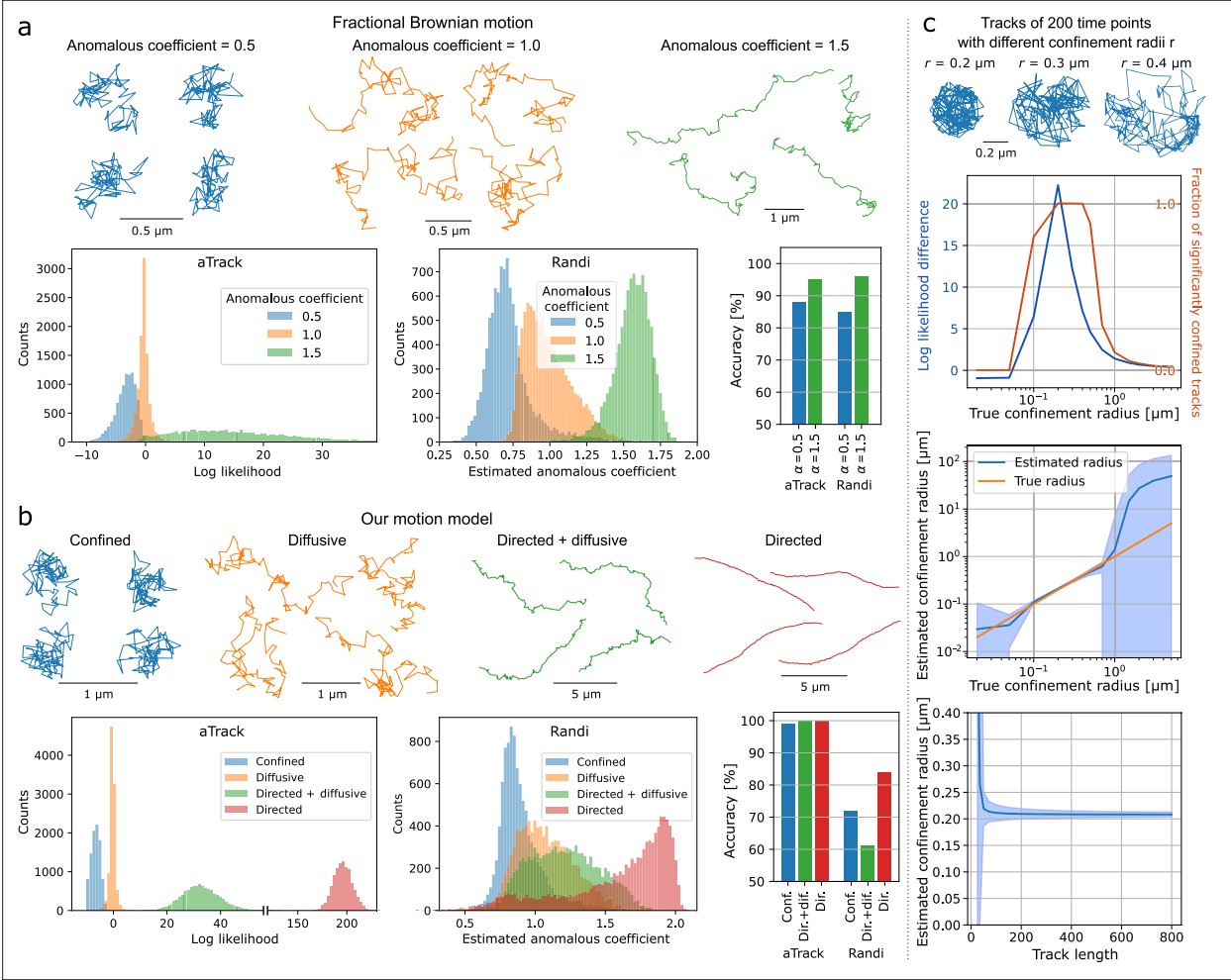

**Figure 6.** Model robustness with other motion types. (**a-b**) Example tracks and corresponding distributions used to determine the type of motion for aTrack and Randi (***Argun et al., 2021***). aTrack uses the difference between the likelihood assuming super-diffusion and the likelihood assuming sub-diffusion (bottom-left). To classify tracks using Randi, we used the estimated anomalous exponent. The accuracy is the fraction of correctly labeled tracks in a data set composed of 5000 sub-diffusive or super-diffusive tracks and 5000 Brownian tracks. Classifications were done using the thresholds that best divide the distributions. (**a**) Analysis of tracks with 100 time steps following fractional Brownian motion with anomalous exponent of 0.5 (sub-diffusive), 1 (diffusive), and 1.5 (super-diffusive). (**b**) Analysis of tracks with 100 time steps following our motion model. Confined tracks: diffusion length d = 0.1 μm, localization error σ = 0.02 μm, confinement force $l = 0.2$, fixed potential well. Brownian tracks: $d = 0.1$ μm, σ = 0.02 μm. tracks in both directed and diffusive motion: $d = 0.1$ μm, σ = 0.02μm , directional velocity $v$ = 0.1 μm·Δt$^{-1}$. Directed tracks: d = 0. μm, σ = 0.02 μm , $v$ = 0.1 μm·Δt$^{-1}$, angular diffusion coefficient 0.1 Rad$^2$s$^{-1}$. (**c**) Analyzing tracks confined by hard boundaries using aTrack. A simulated track with 200 time points diffusing on disks of different sizes. Top panel: Log likelihood difference $L_c - L_B$ and fraction of significantly confined tracks (likelihood ratio $l_B/l_c < 0.05$) depending on the confinement radius. Middle panel: Estimated confinement radius ($= 3\frac{d}{\sqrt{2l}}$) depending on the true confinement radius. Bottom: estimated confinement radius depending on the track length. Blue areas: standard deviations of the estimates.

The online version of this article includes the following figure supplement(s) for figure 6:

**Figure supplement 1.** Effect of dynamic and static localization error on estimated motion parameters.

## Robustness to model mismatches

One of the most important features of a method is its robustness to deviations from its assumptions. Indeed, experimental tracking data will inevitably not match the model assumptions to some degree, and models need to be resilient to these small deviations. To test the generalizability of our approach to other types of motion, we simulated tracks using a different motion model, namely fractional Brownian motion (***Mandelbrot and Van Ness, 1968***). This was performed with three anomalous diffusion exponents, 0.5, 1.0, and 1.5, corresponding to sub-diffusion, Brownian diffusion, and super-diffusion, respectively (***Figure 6a***). The performance of aTrack was then measured by computing the difference between the likelihood of the directed-motion model and the confined-motion model. This

metric is closely related to the likelihood ratio mentioned earlier and has the advantage of showing all three motion behaviors. We expect the likelihood difference to be <0 for sub-diffusive motion, near 0 for Brownian diffusion, and >0 for super-diffusion. We verified this by plotting a histogram of the log-likelihood differences for each type of motion and estimated the classification accuracy in detecting anomalous diffusion from Brownian diffusion. aTrack was 88% accurate when differentiating sub-diffusive fractional Brownian motion (anomalous diffusion exponent of $\alpha = 0.5$) from Brownian motion ($\alpha = 1$), and 95% accurate for differentiating super-diffusive fractional Brownian motion ($\alpha = 1.5$) from Brownian motion.

To compare our approach to one of the leading methods specifically designed to characterize fractional Brownian motion, we performed the same test with Randi (*Argun et al., 2021*), the best-performing machine learning method in a head-to-head comparison of available techniques (*Muñoz-Gil et al., 2021*). On the same fractional Brownian motion dataset, Randi and aTrack achieved similar accuracies.

We then used the same approach to compare aTrack and Randi on tracks generated with our motion model, which differs from Randi's anomalous motion models (*Figure 6b*). Here, we created a dataset of anomalous diffusion with parameters that result in tracks with qualitatively similar properties and MSD curves to those observed with fractional Brownian motion (see tracks in *Figure 6a&b*). For these data, our model performs with high accuracy, at least 99%. In contrast, Randi shows a lower classification accuracy. In particular, Randi has difficulty differentiating diffusive tracks from tracks with both diffusion and directed motion, 61% accuracy (only 11% better than random labeling). Curiously, directed versus Brownian tracks were also surprisingly inaccurate (83%), considering the striking difference in track behaviors (see *Figure 6b*, diffusive versus directed examples).

To test another type of mismatch between our hidden variable model assumptions, we simulated diffusive tracks confined within rigid boundaries. This differs from our model, which uses a potential well to model confinement. We varied the radius of the rigid boundary for simulated tracks and measured the effect on the estimated confinement radius (*Figure 6c*). aTrack accurately determines the confinement radius for a wide range of confinement radii, where the calculated confinement radius is estimated as three times the standard deviation of the potential well. The lower radius bound of the operating range relates to the diffusion length per step, while the upper bound is limited by the extent of the particle's exploration of the confinement area. Note that the exploration distance is determined by the track length and by the diffusion length.

Motion blur is another experimental effect in single-particle tracking that can bias parameter estimation. While our model does not explicitly account for it, the estimated diffusion coefficient can be easily corrected to adjust for that effect (*Figure 6—figure supplement 1*). As described by *Berglund, 2010*, static and dynamic localization errors have antagonistic effects on the offset term of the MSD. Our model, which explicitly models static localization error but not dynamic error, yields good estimates of the diffusion length if $d < \sqrt{6}\sigma$ (MSD curve with a positive offset), but it underestimates $d$ by a factor $\sqrt{2/3} \approx 0.82$ if $d > \sqrt{6}\sigma$. To explicitly adapt our tool to motion blur, one can include motion blur using our new framework for model design (*Simon and Cardona, 2025*).

## Implementation on experimental data

To test the usefulness of aTrack on experimental data, we performed classification, population characterization, and parameter-estimation experiments.

First, we applied aTrack to analyze the movement of the spindle pole body (SPB) in *Saccharomyces cerevisiae* (*Figure 7a*). The SPB is the microtubule-organizing center in yeast and is embedded in the nuclear envelope. Directed motion of the SPB occurs during S phase, where actin is required to establish spindle orientation by directing the spindle pole body towards the bud neck as well as during mitosis when the spindle elongates to separate the chromosome masses of mother and daughter cell (*Theesfeld et al., 1999*). To visualize SPB dynamics, we imaged unsynchronized cells expressing Spc42-mCherry at a time resolution of 100ms and used treatment with Latrunculin A (Lat A) to disrupt actin polymerization. Computing the MSD of the population of tracks showed that, on average, tracks appear diffusive (*Figure 7—figure supplement 1a*). To go beyond this ensemble metric, we used aTrack to compute the likelihood ratio ($l_b/l_d$) of each track of 100 time points and used this metric to classify tracks as significantly directed or not with a type 1 error rate of 5% (*Figure 7b*). Then, we computed the fractions of directed tracks (*Figure 7c*). In untreated cells, 23% of the tracks exhibited

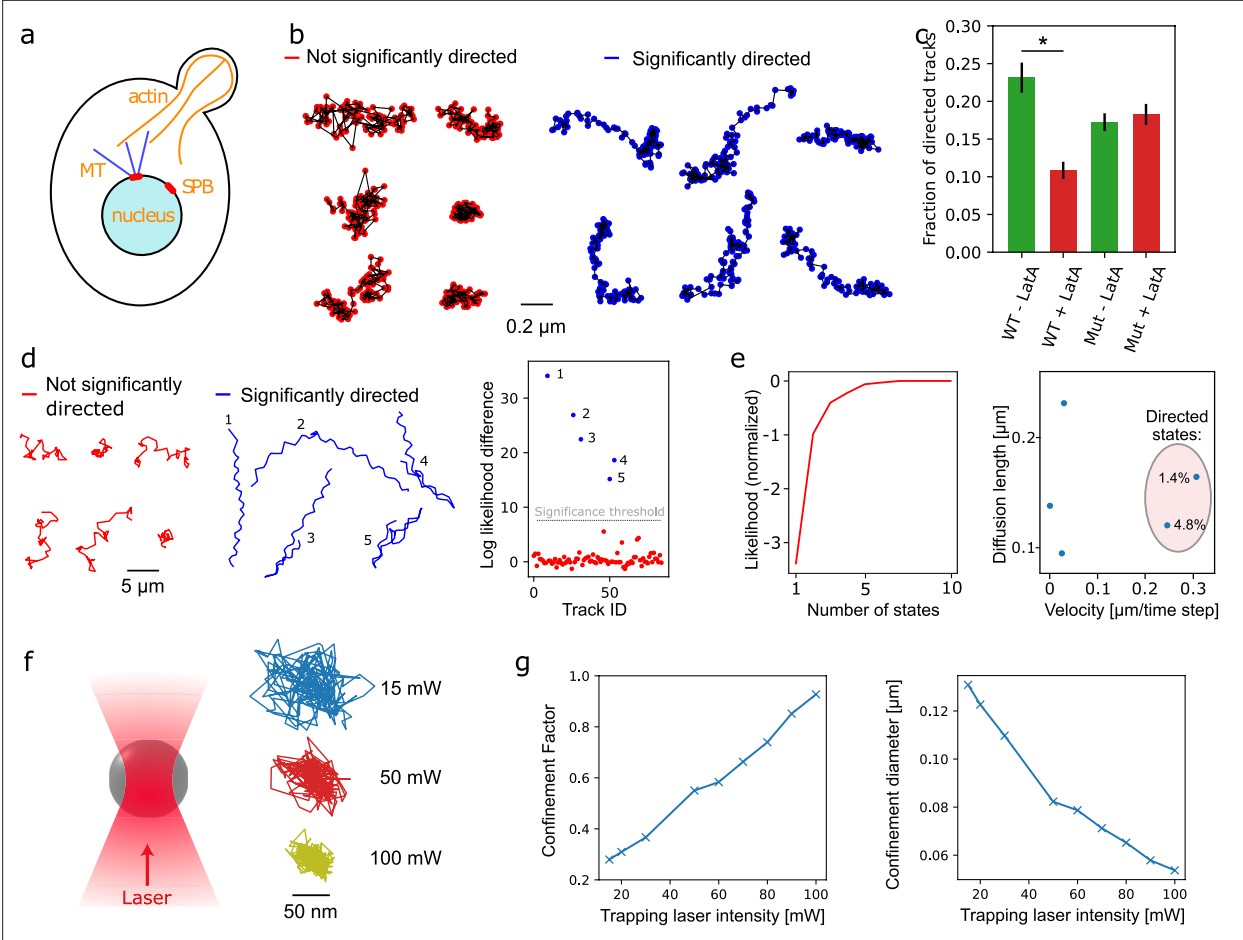

**Figure 7.** Experimental demonstrations. (**a**) Illustration showing the interaction of the budding yeast spindle pole body (SPB) with actin via microtubules (MT). Actin-dependent motors are responsible for moving the nucleus toward the bud neck during S-phase. (**b**-**c**) Analysis of spindle pole body (SPB) tracks. The analysis was carried out on tracks of 100 time points. (**b**) Examples of tracks classified by aTrack to be either significantly directed or non-significantly directed. A random selection of tracks colored by their associated likelihood ratios with and without LatA treatment can be found in *Figure 7—figure supplement 1b*. (**c**) Mean fraction of directed tracks from three biological replicates for the WT and two biological replicates for the latrunculin-resistant mutant. Each replicate contains at least 640 tracks for an average of 4180 tracks per replicate. Error bars: standard deviation. *: significant difference according to a t-test with independent variables (p-value = 0.00165). (**d**-**e**) Analysis of gold nanoparticle (NP) tracks in the presence of motile bacteria (50 time points per track), where some NPs adhere to cells. (**d**) NP tracks colored according to their state of motion classification using aTrack's single-track statistical test and the log likelihood difference ($L_d - L_b$) of all tracks. Tracks are considered significantly directed if the likelihood ratio (which is an overestimate of the p-value) is lower than 0.05 (type I error) divided by the number of tracks (85) according to the Bonferroni correction (=log likelihood difference > 7.44). (**e**) Maximum likelihood (per track) of the population of tracks depending on the number of states (minus the likelihood assuming 10 states). (**f**-**g**) Analysis of tracks for 1 µm beads trapped using optical tweezers with different laser powers. (**f**) Illustration of the optical trap and example tracks of 100 time points for different laser powers. (**g**) Fitting a single-state confined diffusion model on a population of 300 tracks with 20 time points.

The online version of this article includes the following figure supplement(s) for figure 7:

**Figure supplement 1.** Confinement characterization in *Saccharomyces cerevisiae*.

significant directed motion. In contrast, cells treated with LatA showed significantly lower fractions of directed tracks (10%, p-value = 0.00165). This drop in the fraction of directed tracks was not observed in a LatA-resistant strain. Thus, aTrack can reliably detect directed actin-dependent motion at timescales of 10 s. See *Figure 7—figure supplement 1a* for a wider selection of tracks and their associated classifications.

To test the applicability of aTrack to the field of biosensing, we performed a tracking experiment with highly visible gold nanoparticles (AuNP) and *E. coli*. Specifically engineered gold nanoparticles (AuNPs) can attach to cells, and many species are motile (*Zapata-Farfan et al., 2023*). Thus, detecting a directed fraction of AuNPs could be used for sensitive and fast biodetection in complex

environments. AuNPs were added to a diluted *E. coli* culture and imaged. While free AuNPs diffuse rapidly, our method identified directed tracks in a 5.9% population with very high certainty (likelihood ratio ¡ 0.001, *Figure 7d*). This directed fraction is readily apparent by manual data inspection (see supplemental video). Notably, while the directed fraction also exhibited lateral oscillations, consistent with cell rotation (*Powers, 2002*), our model was robust to this deviation from the model assumptions. Next, we analyzed the tracks at the population level and computed the number of states as a function of likelihood (*Figure 7e*). The likelihood function shows a noticeable difference between four and five states. In the five-state model, there are two directed states and three diffusive states. The diffusive states likely represent free AuNPs as well as NPs conjugated to diffusing debris of different sizes. The two directed populations are likely caused by the abrupt tumbling motion present in some tracks. These directed states comprised 6.2% of tracks, similar to the analysis of individual tracks.

Notably, in this dataset, the likelihood plot shows an ambiguous number of states, where the likelihood increases marginally after five states. Such ambiguities are expected when the dataset and the model assumptions do not perfectly match. To ensure that the directed fraction is well estimated independently of the number of states, we varied the number of states to see how the parameter estimates changed, as in *Prindle et al., 2022*. We found that using three to seven states resulted in the same fraction of clearly directed tracks with similar parameters.

Finally, we tested our method's capability to detect confinement. To do so, we confined 1 μm beads in an optical trap (*Ashkin and Dziedzic, 1987*) and varied the laser power to control the trap stiffness. Based on populations of 300 tracks of 20 time points, we measured the confinement factor $l$ and confirmed that it increases with the laser power of the optical trap, while the calculated confinement diameter $u$ decreases, where $u = \frac{4d}{\sqrt{2l}}$ (*Figure 7g*).

## Discussion

aTrack is a new tool for classifying and characterizing noisy tracks, which performs well in a diverse range of conditions. Specifically, the framework's flexibility is relevant for a wide variety of diffusive, confined, and directed motion types. Our tool classifies the motion type as Brownian or not and quantifies biologically relevant motion parameters, such as the diffusion coefficient, confinement diameter, and velocities. Importantly, this approach calculates how statistically robust a classification is regarding the likelihood difference from a Brownian diffusion model. The flexibility of our approach in capturing a variety of motion-type behaviors has the usefulness of the catch-all approach of using an anomalous exponent. However, unlike the anomalous exponent that fits multiple underlying motion models (*Metzler et al., 2014*), our framework has the advantage of outputting interpretable parameters that describe the motion, for example velocity or confinement radii.

aTrack has several features that make it advantageous for analyzing directed and confined motion. For example, allowing the anomalous variable to change over time increases flexibility. For directed tracks in cells, motion is rarely exactly straight over long distances. Allowing changes in velocity makes it possible to classify and characterize these curved trajectories robustly. Analogously for confined motion, a confined particle may leave its local environment or hop between environments (*Golding and Cox, 2006*; *Weigel et al., 2011*; *Ehrig et al., 2011*). Our model accounts for this by allowing the center of the confinement well to diffuse as well. Interestingly, we found that directed motion can be readily identified from very short tracks due to the deterministic nature of this type of motion. Longer tracks are needed to classify confined motion with the same statistical certainty. This is to be expected, as a confined particle should reach the boundary of the confinement area several times to be distinguishable from a freely diffusive particle.

As with all classification tools, a source of ambiguity arises when motion types resemble one another. For example, this confusion occurs between immobile particles and very tightly confined particles, which can sometimes be indistinguishable and not necessarily insightful. Indeed, immobilized fluorophores can be modeled as confined to an area around a substrate to which they are bound. To better distinguish these ambiguous behaviors, experimental modifications are needed, such as adapting the experimental framerate or improving the localization precision using brighter fluorophores (*Mickolajczyk and Hancock, 2017*; *Simon et al., 2024*).

While machine-learning tools have proven to be effective for sub-diffusion characterization (*Muñoz-Gil et al., 2021*), our approach shows it is possible to achieve similarly high performance with many fewer parameters. For our tool, these parameters map onto the stochastic physical behaviors

of particle motion, making interpretation more straightforward. Compared to the machine-learning tool we tested, we found that aTrack is more robust to small model mismatches. This finding is consistent with the well-documented issues of machine-learning models generalizing poorly to new data. Of course, in the context of tracking, the generalizability of a model to new data is a key factor, as experimentally obtained data never perfectly match the model assumptions or the training data set. One solution to make a machine learning model that generalizes better is to use a physics-informed neural network (**Raissi et al., 2019**). Such a network would use probabilistic relationships to efficiently learn the physical properties of unlabeled tracks and would contain far fewer parameters than classical networks.

It is often useful to assume a finite number of states with fixed parameters to model the various molecular states in a sample (**Simon et al., 2024**). Selecting the right number of states is difficult, but can be automated by different methods. The criterion, for example AIC and BIC, used to determine the number of states is reliable when the model and the data are in perfect agreement. However, experimental data never match the model assumptions perfectly, and even discrepancies between models and simulated data can affect reliability, such as continuous-time simulations and discrete models. This usually results in overestimating the number of states for large data sets (**Lindén and Elf, 2018**). To avoid this flaw of classical criteria, we showed that a penalization term proportional to the number of states and to the log likelihood of the data can prevent the spurious increase of the number of states when increasing the number of tracks. The drawback of this approach is the addition of a tunable parameter that influences the number of estimated parameters of the model, and it may still be necessary to limit the number of states to the biologically relevant system, or consider groups of states, for example all significantly directed states.

An important limitation of our approach is that it presumes that a given track follows a unique underlying model with fixed parameters. In biological systems, particles often transition from one motion type to another; for example, a diffusive particle can bind to a static substrate or molecular motor (**Weiss et al., 2023**). In such cases, or in cases of significant mislinkings, our model is not suitable. However, this limitation can be alleviated by implicitly allowing state transitions with a hidden Markov Model (**Simon et al., 2023**) or alternatives such as change-point approaches (**Briane et al., 2020**; **Argun et al., 2021**; **Manzo, 2021**) spatial approaches (**Türkcan and Masson, 2013**).

In conclusion, we have shown that aTrack can identify anomalous diffusion and parameterize the motion over a broad range of motion types using a robust probabilistic framework. As the motion-model parameters estimated by the method represent physical phenomena, these variables are readily interpretable, for example the diffusion coefficient, confinement radius, and velocities. Finally, the employed motion models were selected to permit analytical integration, which makes calculating the model parameters fast and accurate; of course, this integration strategy can be implemented for a variety of motion models with hidden states, further expanding the applicability of this approach to other motion types.

## Methods
### Modeling particle motion with observed and hidden variables
#### Probabilistic model for confined motion
At each time step $i$, our confined-motion model consists of (1) a Brownian motion step, (2) a confinement step, (3) an update to parameters, and (4) a localization error step. The Brownian motion step updates the particle's position $r_i$ to an intermediate position $z_i$. The variable $z_i - r_i$ follows a Gaussian distribution centered at 0 with standard deviation $d$, where $d$ is the diffusion length, $d = \sqrt{2D\Delta t}$, $D$ is the diffusion coefficient, and $\Delta t$ is the time step. Next, the confinement step is modeled by an attractive force between the particle and a potential well centered at $h_i$. More precisely, the particle moves toward the center of the potential well proportionally to a confinement factor $l$ and to the distance $r_i - h_i$. The next real position $r_{i+1}$ is thus determined by the following relationship $r_{i+1} = (1 - l) \cdot z_i + l \cdot h_i$. To allow the potential well to move, $h$ is updated at each step such that $h_{i+1} - h_i$ follows a Gaussian distribution of mean 0 and standard deviation $q$. Finally, to model localization error, the observed positions $c_i$ and the real positions $r_i$ are related by the localization precision, $\sigma$, where $c_i - r_i$ follows a Gaussian distribution of mean 0 and standard deviation $\sigma$.

The distribution of positions for a diffusive particle in a fixed potential well is a Gaussian distribution with standard deviation $\rho = \sqrt{\frac{D \cdot dt}{l}} = \frac{d}{\sqrt{2l}}$. As we have no prior information about the initial center of the potential well, we assume it is positioned according to a Gaussian probability density function centered on the initial observed position. While it would be even better to consider a Gaussian $f_{h_0}$ centered around $r_0$, we simplify it by approximating it to be centered around $c_0$ and of standard deviation $q_0 = \sqrt{\rho^2 + \sigma^2}$. The joint probability density function corresponding to this model is the product of Gaussian functions shown in **Equation 1**, which is integrated over all hidden parameters to calculate the likelihood, $l_{confined}$:

$$
\begin{aligned}
&f_\sigma(r_0 - c_0) f_d(z_0 - r_0) f_{q_0}(h_0 - c_0) \\
&f_\sigma((1-l)z_0 + lh_0 - c_1) f_d((1-l)z_0 + lh_0 - z_1) f_q(h_1 - h_0) \\
&f_\sigma((1-l)z_1 + lh_1 - c_2) f_d((1-l)z_1 + lh_1 - z_2) f_q(h_2 - h_1) \\
&[\ldots] \\
&f_\sigma((1-l)z_{n-2} + lh_{n-2} - c_{n-1}) f_d((1-l)z_{n-2} + lh_{n-2} - z_{n-1}) f_q(h_{n-1} - h_{n-2}) \\
&f_\sigma((1-l)z_{n-1} + lh_{n-1} - c_n)
\end{aligned}
\tag{1}
$$

with the real positions of the particle $r_i$, potential well $h_i$, and intermediate position $z_i$ are hidden variables, and $l$ is the confinement factor. By integrating this joint probability over the hidden variables, we can retrieve the probability of the track (the observed positions) given the model and its parameters.

While this integration step can be computed with a Monte Carlo approach (Slator and Burroughs, 2018), this is computationally expensive. Instead, we integrate using an analytical recurrence formula. This formula is allowed by the fact that the joint probability density function is a product of Gaussians and by the property that for two Gaussians, $f$ and $g$, with means $\mu_f$, $\mu_g$ and standard deviations $\sigma_f$, $\sigma_g$, the product is also Gaussian, $f(x) \cdot g(x) = \phi \cdot \eta(x)$ where $\phi$ and $\eta$ are Gaussian distributions described by **Equation 2**.

$$
\phi = \frac{\exp\left(-\frac{(\mu_f - \mu_g)^2}{2 \cdot (\sigma_f^2 + \sigma_g^2)}\right)}{\sqrt{2\pi (\sigma_f^2 + \sigma_g^2)}} \text{ and } \eta(x) \text{ a Gaussian of mean } \mu_{fg} = \frac{\sigma_f^2 \mu_g + \sigma_g^2 \mu_f}{\sigma_f^2 + \sigma_g^2} \text{ and variance } \sigma_{fg}^2 = \frac{\sigma_f^2 \sigma_g^2}{\sigma_f^2 + \sigma_g^2}. \tag{2}
$$

See Supplementary information for more details.

## Probabilistic model for directed motion

To consider directed motion, we use the same general framework as confinement. At each time step $i$, the real particle position $r_i$ is first updated to an intermediate position $z_i$, where the variable $z_i - r_i$ follows a Gaussian distribution centered at 0 with standard deviation $d$. Next, we add the directed-motion component with a vector $w_i$, such that the next real position is the sum of the two steps, $r_{i+1} = z_i + w_i$. Combining these two substeps, we get $z_i - r_i = r_{i+1} - r_i - w_i$, simplifying the integration process of the probability density function expressed in **Equation 3**. Analogous to our confinement model, the velocity vector, $w_i$, is allowed to change over time, where $w_{i+1} - w_i$ follows a Gaussian distribution with mean 0 and standard deviation $q$. Finally, we include the effect of localization error, where the observed position $c_i$ is related to the real position $r_i$ following a Gaussian distribution, where $c_i - r_i$ is Gaussian distributed with mean 0 and standard deviation $\sigma$.

During the first time step, the orientation and length of the directed motion vector (speed) need to be initialized. To do so, we assume $w_0$ follows a Gaussian distribution function with mean 0 and standard deviation $v$. The resulting joint probability density function is shown in **Equation 3**.

$$f_\sigma(r_0 - c_0) f_v(w_0) f_d(r_1 - r_0 - w_0) f_q(w_1 - w_0)$$

$$f_\sigma(r_1 - c_1) f_d(r_2 - r_1 - w_1) f_q(w_2 - w_1)$$

$$f_\sigma(r_2 - c_2) f_d(r_3 - r_2 - w_2) f_q(w_3 - w_2)$$

$$[...] \tag{3}$$

$$f_\sigma(r_{n-2} - c_{n-2}) f_d(r_{n-1} - r_{n-2} - w_{n-2}) f_q(w_{n-1} - w_{n-2})$$

$$f_\sigma(r_{n-1} - c_{n-1}) f_d(r_n - r_{n-1} - w_{n-1})$$

$$f_\sigma(r_n - c_n)$$

As with confinement, the probability of a directed track given the model parameters can be calculated using an analytical recurrence formula to integrate over the hidden positions, $r_{0 \to n}$, and velocities $w_{0 \to n-1}$. The parameter $v$ can, in principle, be used to estimate the velocity of a particle with a constant speed but changing orientation in the imaging plane; however, to estimate the average speed of a particle, we use another metric, $k$, that appears in our integration process (see Supplementary information).

## Modeling time-varying velocities and changes in direction

In our model, each axis, x and y, is treated separately, and the velocities can evolve according to a Gaussian distribution. This allows a particle's direction to change over time. To quantify how direction changes affect the analysis for particles with a constant speed, we simulated tracks with a fixed speed and time-dependent direction for a range of angular diffusion coefficients, $D_\theta$. The model parameter, $q$, represents the standard deviation of the change in speed, which can be converted to an angular diffusion length using the following trigonometric relation and the estimated speed of the motion.

In the case of pure directed motion with direction changes, let us consider a single time step, $i$ and the following geometric construction. We have A, the previous particle position $r_{i-1}$; B, the particle position assuming no changes of orientation or diffusion, $r_{i-1} + v_{i-1}$; and C, the actual particle position after a change of orientation, $r_i$. ABC forms an isosceles triangle, which can be split into two right triangles. We find the following relationship between the scalar distances BC, AB, and $\theta$, $sin(\frac{\theta}{2}) = \frac{BC/2}{AB}$, where $\theta$ is the orientation angle change.

## Fitting method

For the pure Brownian model, the parameters are the diffusion coefficient and the localization error. For the confinement model, the parameters are the diffusion coefficient, the localization error, confinement factor, and the diffusion coefficient of the potential well. For the directed model, the parameters are the diffusion coefficient, the localization error, the initial velocity, and the acceleration variance.

These parameters are estimated using the maximum likelihood approach, which consists of finding the parameters that maximize the likelihood. We realize this fitting step using gradient descent via a TensorFlow model. All the estimates presented in this article are obtained from a single set of initial parameters to demonstrate that the convergence capacity of aTrack is robust to the initial parameter values.

## Multi-state population model

We designed a specific algorithm to retrieve the number of states in a data set and estimate the parameters of each state as described in the code of the script *atrack.py* available on our Github page https://github.com/FrancoisSimon/aTrack, (**Simon, 2026**). This multi-state population algorithm starts by performing individual fittings of tracks to get a type of motion and a set of parameters for each track. Then, we use a Gaussian mixture model on the parameters of the individual tracks to provide an overestimate of the number of states (e.g. 20 if we expect 5 states) and fit the model so that every actual state underlying the data is well represented by at least one of the model states. Next, we iteratively remove the least useful model state and refit the model until we obtain a single-state model. At each iteration, the least useful state is determined as the state whose removal has the smallest negative impact on the likelihood.

## Track simulations

We subdivided each time step into 20 substeps to simulate approximately continuous tracks. We applied the Brownian diffusion and anomalous movements to each of these substeps. For confined motion, the latter step moves the particle toward the center of the potential well center proportionally to the distance multiplied by a scaled confinement factor, $l/20$. In simulations with a moving potential well, the center moves according to the well's diffusion coefficient, $q$. For directed motion, we applied a shift of constant velocity at each time step after the diffusion step. The orientation of the directed motion could vary according to a rotational diffusion coefficient. The particle's position after the 20 substeps was set to the particle's real position, $r_i$, and the localization error was added to create the observed position $c_i$. This process is equivalent for undivided frames for directed motion, as the diffusion and the directed motion steps are independent. However, for the confined model, where the diffusion of the particle influences the anomalous step, thus there is a dependence between the diffusion and confinement steps.

Fractional Brownian motion was simulated using the Python package 'fbm' (https://pypi.org/project/fbm/) with the *Davies and Harte, 1987*.

## Experimental methods

### Spindle-pole body

Yeast cells expressing Spc42-mCherry with or without allelic mutation in actin act1-113 (*Ayscough et al., 1997*; strains KWY10722 and KWY10328 described in *Zsok et al., 2023*) were grown to exponential growth phase in synthetic complete medium with glucose (SCD). Cells were treated with 0.2 mM Latrunculin A (Enzo Life Sciences, BML-T119-0500) and DMSO as solvent control for 10 min prior to imaging. Matrical 384-well glass bottom plates coated with Concanavalin A were used to image treated cells on a temperature-controlled inverted Nipkow spinning disk microscope equipped with the Yokogawa Confocal Scanner Unit CSU-W1-T2 controlled by the VisiVIEW Software (Visitron). It was used in spinning disk mode with a pinhole diameter of 50 µm combined with a 1.45 NA, 100 x objective. Images were acquired on an EMCCD Andor iXon Ultra camera (1024x1024 pixel, 13x13 µm pixel size); for one of three biological replicates, the data was acquired with dual camera settings. Imaging was performed at 30 °C with 80% laser intensity of a Diode 561 nm, 200 mW laser. Timelapse data were acquired with 100ms exposure time in stream mode for 300 frames. For all movies, tracks were obtained using the ImageJ (*Schneider et al., 2012*) plugin (*Tinevez et al., 2017*). Peak detection: LoG with radius = 0.45 µm and threshold = 20, linkage: simple lap tracker with a linking maximum distance of 0.5 µm and allowing gaps of one frame. In order to account for the abrupt changes of motion, the fractions of directed tracks were inferred from segments of 20 time steps. More precisely, the log likelihood of each segment was computed assuming either Brownian motion or directed motion, and for each hypothesis, the log likelihoods of the segments were summed. The resulting log likelihoods were used to compute the log likelihood ratio for each track of 100 time points. Then, we applied a 5% threshold to classify tracks as directed or not. See *Figure 7—figure supplement 1b* to visualize a wide random selection of tracks classified according to this procedure.

### Bacteria detection and tracking

Bacteria detection with nanoparticles was performed following the protocol described in *Zapata-Farfan et al., 2023*. In brief, *E. coli* (strain 25922, ATCC) were cultured for 24 hr in Trypticase soy broth (TSB) at 37 C and then incubated for 30 min with 100 nm spherical gold nanoparticles at 50 µg/mL (A11-100-CIT, Nanopartz) in 1 x PBS. Samples were placed in a small chamber consisting of double-sided tape (5 µm thickness, Nitto) between a coverslip and glass slide. Tracking was performed using Trackmate 7.0 *Tinevez et al., 2017* with the LoG detection method with a diameter of 0.70 µm (6 pixels) and quality threshold of 24.

### Optical tweezers

The tracking of microspheres captured in an optical trap was achieved using a custom instrument constructed on a modular inverted microscope (MIM/RAMM, ASI Imaging), incorporating optical trapping and imaging paths. The trapping diode laser (SNP-06E-100, Teem) emitting at 1064 nm (TEM$_{00}$) had an average output power of 60 milliwatts. The laser beam is expanded using a 1:3 telescope

(Achromatic doublets, Thorlabs) to overfill the objective's back focal plane. This expanded beam was focused through a 100 X, 1.45 NA objective (UPlanXApo 100 X, Olympus) to trap dielectric particles. The same objective was used to collect bright-field illumination, which was imaged using a CMOS camera (ORCA - Flash4.0 LT3, Hamamatsu). A three-axis piezo-driven stage (MicroScan SCXYZ100, Thorlabs) with a precision of 25 nm facilitated sample movement to capture particles in the trap. One µm polystyrene microbeads (Monodisperse fluorescent microspheres, Cromtech Research Center) were suspended in water. The solution was squeezed between two coverslips (No. 1.5, Thermo), and a single bead was brought into the laser trap by translating the sample. The intensity of the laser trap was varied from 15 mW to 100 mW. The 40 mW movie was excluded as we found it to be out of focus. Data acquisition was performed at 315 frames per second for 2 min. Bead positions were tracked using (*Tinevez et al., 2017*). aTrack analysis was performed on 300 segments of 20 time points, inputting a fixed diffusion coefficient, which were determined from the lowest laser intensity (15 mW) $d = 0.028$ µm, respectively. Confinement radius = $4d/\sqrt{2l}$.

## Acknowledgements

The authors thank Sven van Teeffelen for helpful discussions. This work was supported by the Natural Sciences and Engineering Research Council of Canada [NSERC Discovery grant to MM, (RGPIN-06404–2016) to CB, (RGPIN-2022–05142) to LEW], the Canada First Research Excellence Fund (Trans-MedTech Institute), the ETH research grant ETH-33 19–1 and the Swiss National Science Foundation (project number 320030–236124) to ED.

## Additional information

### Funding

| Funder | Grant reference number | Author |
| --- | --- | --- |
| Natural Sciences and Engineering Research Council of Canada | | Michel Meunier |
| Natural Sciences and Engineering Research Council of Canada | RGPIN-06404-2016 | Caroline Boudoux |
| Natural Sciences and Engineering Research Council of Canada | RGPIN-2022-05142 | Lucien E Weiss |
| Swiss National Science Foundation | 320030-236124 | Elisa Dultz |
| Eidgenössische Technische Hochschule Zürich | ETH-33 19-1 | Elisa Dultz |
| Canada First Research Excellence Fund | TransMedTech Startup Funds | Lucien E Weiss |

The funders had no role in study design, data collection and interpretation, or the decision to submit the work for publication.

### Author contributions

François Simon, Conceptualization, Data curation, Software, Formal analysis, Validation, Investigation, Visualization, Methodology, Writing – original draft, Project administration, Writing – review and editing; Guillaume Ramadier, Investigation, Visualization, Methodology, Writing – original draft, Writing – review and editing; Inès Fonquernie, Software, Investigation, Writing – review and editing; Janka Zsok, Validation, Investigation, Visualization, Methodology, Writing – review and editing; Sergiy Patskovsky, Data curation, Investigation, Methodology, Writing – review and editing; Michel Meunier, Caroline Boudoux, Supervision, Funding acquisition, Writing – review and editing; Elisa Dultz, Supervision, Funding acquisition, Validation, Investigation, Visualization, Methodology, Writing – original draft, Writing – review and editing; Lucien E Weiss, Conceptualization, Supervision, Funding

acquisition, Visualization, Methodology, Writing – original draft, Project administration, Writing – review and editing

## Author ORCIDs
François Simon (ID) https://orcid.org/0000-0002-0737-2312
Guillaume Ramadier (ID) https://orcid.org/0000-0002-5175-1643
Sergiy Patskovsky (ID) https://orcid.org/0000-0001-7175-481X
Michel Meunier (ID) https://orcid.org/0000-0002-2398-5602
Caroline Boudoux (ID) https://orcid.org/0000-0002-2145-9086
Elisa Dultz (ID) https://orcid.org/0000-0003-1114-5523
Lucien E Weiss (ID) https://orcid.org/0000-0002-0971-7329

Reviewer #1 (Public review): https://doi.org/10.7554/eLife.99347.3.sa1
Reviewer #2 (Public review): https://doi.org/10.7554/eLife.99347.3.sa2
Author response https://doi.org/10.7554/eLife.99347.3.sa3

## Additional files

### Supplementary files
MDAR checklist

### Data availability
aTrack is available as a stand-alone software for Windows and as a python package on Github https://github.com/FrancoisSimon/aTrack, copy archived at *Simon, 2026*. The Github page also provides a tutorial, and installation instructions. The datasets are available on Zenodo https://doi.org/10.5281/zenodo.18420221.

The following previously published dataset was used:

| Author(s) | Year | Dataset title | Dataset URL | Database and Identifier |
|---|---|---|---|---|
| Simon F, Ramadier G, Zsok J, Patskovsky S, Dultz E, Weiss LE | 2026 | Dataset used for the aTrack article | https://doi.org/10.5281/zenodo.18420221 | Zenodo, 10.5281/zenodo.18420221 |

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

# Appendix 1

## Statistical tests

Statistical tests are important as they allow one model to be chosen over another. In the case of independent and identically distributed variables, the log likelihood ratio of a dataset drawn from the null hypothesis follows a chi-squared distribution that can be used to determine the p-value. The *p*-value can then be used to reject or not reject the null hypothesis.

### Bounding the p-value with the likelihood ratio

For a given trajectory $\mathbf{c} = \{c_0, c_1, \ldots, c_n\}$, we construct likelihood ratio tests comparing a null hypothesis $H_0$ of Brownian diffusion against an alternative hypothesis of $H_1$ anomalous diffusion. (confined or directed). The likelihood ratio is defined as $\Lambda\left(\mathbf{c}\right) = \frac{L\left(\hat{\theta}_0|\mathbf{c}\right)}{L\left(\hat{\theta}_1|\mathbf{c}\right)}$, where $L\left(\hat{\theta}_i|\mathbf{c}\right)$ denotes the maximum likelihood underhypothesis $H_i$ with maximum likelihood estimator $\hat{\theta}_i$.

While under standard regularity conditions, Wilks' theorem (***Wilks, 1938***) establishes that $-2\log\Lambda(c)$ asymptotically follows a $\chi^2$ distribution with degrees of freedom equal to the difference in parameter dimensions between models, this theorem cannot be rigorously applied here as the null hypothesis assumes that the anomalous parameter lies at the boundary of its parameter space (under $H_0$, the anomalous variable follows a Dirac delta function) (***Chen et al., 2020***; ***Van der Vaart, 2000***). Through simulation (***Figure 2—figure supplement 2***), we empirically found that under $H_0$, $\Lambda(c)$ consistently concentrated near 1 (track lengths were tested from 5 to 400 time points). This shows that under these conditions, the likelihood ratio $\Lambda(c)$ satisfies $P(\Lambda(c) \leq \alpha|H_0) \leq \alpha$ As the p-value is by definition uniform and as we can define a bijection between the p-value and the skewed $\Lambda(c)$ for a desired Type I error rate $\alpha$ we can reject $H_0$ when $\Lambda(c) < \alpha$. This guarantees $P(\text{rejects}\,H_0|H_0\text{true}) \leq \alpha$. Since the Brownian model is nested within both confined and directed motion models with confinement factor $l = 0$ or velocity $v = 0$ respectively, we have $L(\hat{\theta}_0|c) \leq L(\hat{\theta}_1|c)$ which ensures $0 \leq \Lambda(c) \leq 1$.

### Simulation-based p-value determination

While easy to implement, we acknowledge that the above method has two flaws: (1) it is based on the empirical observation that the $\Lambda(c)$ distribution is skewed towards 1. This may not hold for tracks outside the tested range, for example those with more than 400 data points. (2) Using an inequality instead of realizing the bijection between $\Lambda(c)$ and the p-value lacks power. The more $\Lambda(c)$ under $H_0$ concentrates near 1, the greater is this lack of power. Therefore, for applications requiring more precise p-value estimates, a simulation-based calibration procedure can be performed. Given an observed trajectory c with test statistic $\Lambda_{\text{obs}}$, we first estimate null model parameters as $\hat{\theta}_0 = \arg\max_{\theta_0} L(\theta_0|c)$. We then simulate $N$ trajectories under $H_0$ with parameters $\hat{\theta}_0$ and compute the likelihood ratios $\left\{\Lambda^{(i)}\right\}_{i=1}^{N}$. The p-value is estimated as $\hat{p} = \frac{1}{N}\sum_{i=1}^{N}[\Lambda^{(i)} \leq \Lambda_{\text{obs}}]$. By the law of large numbers (***Ferguson, 2017***), $\hat{p} \to P(\Lambda(c) \leq \Lambda_{obs}|H_0)$ as $N \to \infty$. When analyzing individual trajectories from a population of tracks, multiple testing corrections become essential to control for false positives. For $m$ trajectories with individual p-values $\left\{p_i\right\}_{i=1}^{m}$ we can apply either the Bonferroni correction (***Bonferroni, 1936***), rejecting $H_0^{(i)}$ if $p_i < \alpha/m$, or the Benjamini-Hochberg procedure (***Benjamini and Hochberg, 1995***) to control false discovery rate at level $\alpha$.

### Multi-model comparisons

The likelihood ratio framework extends to multi-model comparisons. For model selection among $K$ candidates, one can employ information criteria including the Akaike Information Criterion (***Akaike, 1974***) $\text{AIC} = -2\log L(\hat{\theta}) + 2k$, or the Bayesian Information Criterion (***Schwarz, 1978***), $\text{BIC} = -2\log L(\hat{\theta}) + k\log n$, where $k$ is the number of parametersand $n$ is the trajectory length. Our empirical analysis (Figure S??) demonstrates that classical information criteria often overestimate the number of motion states, particularly for large datasets. This occurs due to model misspecification, as experimental data never perfectly match theoretical models (***Burnham and Anderson, 2002***; ***Claeskens and Hjort, 2008***). To address this limitation, we introduce a penalization term proportional

to both the number of parameters and log-likelihood: Modified $\text{BIC} = -2\log L(\hat{\theta}) + k \log n + \beta k L(\hat{\theta})$, where $\beta$ is a small positive constant that prevents spurious state proliferation.

## Derivations

## Likelihood calculation for confined motion

To simplify the integration of the product of Gaussians described in the Method section, **Equation 1**, $f_\sigma((1-l)z_i + lh_i - c_{i+1})f_d((1-l)z_i + lh_i - z_{i+1})$ can be refactored to $K(z_{i+1} - c_{i+1})\, G'((1-l)z_i + lh_i - \frac{\sigma^2}{\sigma^2+d^2}z_{i+1} - \frac{d^2}{\sigma^2+d^2}c_{i+1})$ of variances $k^2 = \sigma^2 + d^2$ and $g'^2 = \sigma^2 d^2/(\sigma^2 + d^2)$. This can be further simplified into $\frac{1}{1-l}K(z_{i+1} - c_{i+1})\, G(z_i + \frac{l}{1-l}h_i - az_{i+l} - bc_{i+1})$ with $a = \frac{\sigma^2}{(1-l)(\sigma^2+d^2)}$, $b = \frac{d^2}{(1-l)(\sigma^2+d^2)}$, $K$ and $G$ represent Gaussian distributions of mean 0 with variances $k^2 = \sigma^2 + d^2$ and $g^2 = \frac{\sigma^2 d^2}{(1-l)(\sigma^2+d^2)}$, respectively. Then, we use the annotation $N(x, V)$ o refer to other Gaussian probability density functions of mean 0 andof variance $V$.

**Equation 1** is equivalent to:

$$f_\sigma(r_0 - c_0)\, f_d(z_0 - r_0)\, f_{h_0}(h_0 - c_0)$$
$$\frac{1}{(1-l)}\, K(z_1 - c_1)\, G(z_0 + \frac{l}{1-l}h_0 - az_1 - bc_1)\, f_q(h_1 - h_0)$$
$$\frac{1}{(1-l)}\, K(z_2 - c_2)\, G(z_1 + \frac{l}{1-l}h_1 - az_2 - bc_2)\, f_q(h_2 - h_1)$$
$$[\ldots]$$
$$\frac{1}{(1-l)}\, K(z_{n-1} - c_{n-1})\, G(z_{n-2} + \frac{l}{1-l}h_{n-2} - az_{n-1} - bc_{n-1})\, f_q(h_{n-1} - h_{n-2})$$
$$\frac{1}{(1-l)}\, f'_\sigma(z_{n-1} + \frac{l}{1-l}h_{n-1} - \frac{c_n}{1-l})$$

$$(4)$$

During the integration process, we obtain the following set of recurrence formulas:
- Initialization:

$$\int \text{over } r_0 : \qquad \int f_\sigma(r_0 - c_0)f_d(z_0 - r_0)\, dr_0 = K(z_0 - c_0)$$

$$\int \text{over } z_0 : \qquad s_0^2 = \left(\frac{1-l}{l}\right)^2 (k^2 + g^2), \quad \zeta_0 = \frac{a}{l}, \quad \eta_0 = \frac{b}{l}c_1 - \frac{1-l}{l}c_0$$

$$\int \text{over } h_0 : \qquad x_0^2 = q_0^2 + q^2, \quad \alpha_0 = \frac{q_0^2}{\zeta_0 x_0^2}, \quad \gamma_0 = \frac{-\eta_0 + \frac{q^2}{x_0^2}c_0}{\zeta_0},$$

$$\qquad\qquad\qquad w_0^2 = \frac{s_0^2 q^2 + q^2 q_0^2 + q_0^2 s_0^2}{\zeta_0^2 x_0^2}, \quad u_0 = c_0$$

$$P_0 \qquad\qquad = \frac{1-l}{l\zeta_0}$$

$$(5)$$

- Recurrence:

$\int$ over $z_i$ :

$$m_i'^2 = k^2 + w_{i-1}^2, \ \tau_i = \frac{l}{1-l} + \frac{k^2}{m_i'^2} \alpha_{i-1}$$

$$s_i^2 = \frac{k^2 \, w_{i-1}^2 + w_{i-1}^2 \, g^2 + g^2 \, k^2}{m_i'^2 \, \tau_i^2}$$

$$\zeta_i = \frac{a}{(1-l) \, \tau_i}$$

$$\eta_i = \frac{1}{\tau_i} \left( \frac{b}{1-l} \, c_{i+1} - \frac{k^2}{m_i'^2} \, \gamma_{i-1} - \frac{w_{i-1}^2}{m_i'^2} \, c_i \right)$$

$$m_i^2 = \frac{m_i'^2}{\alpha_{i-1}^2}$$

$\int$ over $h_i$ :

$$e_i^2 = x_{i-1}^2 + m_i^2$$

$$u_i = \frac{x_{i-1}^2}{\alpha_{i-1} \, e_i^2} \, (c_i - \gamma_{i-1}) + \frac{m_i^2}{e_i^2} \, u_{i-1}, \ \phi_i^2 = \frac{x_{i-1}^2 \, m_i^2}{e_i^2}$$

$$x_i^2 = \phi_i^2 + q^2$$

$$w_i^2 = \frac{\phi_i^2 \, q^2 + q^2 \, s_i^2 + s_i^2 \, \phi_i^2}{x_i^2 \, \zeta_i^2}$$

$$\alpha_i = \frac{\phi_i^2}{x_i^2 \, \zeta_i}, \ \gamma_i = \frac{\frac{q^2}{x_i^2} \, u_i - \eta_i}{\zeta_i}$$

$$P_i = P_{i-1} \cdot \ \frac{1}{\zeta_i \, \alpha_{i-1} \, (1-l) \, \tau_i} \cdot N \left( \frac{c_i - \gamma_{i-1}}{\alpha_{i-1}} - u_{i-1}, \ e_i^2 \right)$$

(6)

- Final step:

$\int$ over $z_{n-1}$ : $\quad m_{n-1}'^2 = k^2 + w_{n-2}^2$

$$m_{n-1}^2 = \frac{m_{n-1}'^2}{\alpha_{n-2}^2}$$

$$\tau_{n-1} = \frac{l}{1-l} + \alpha_{n-2} \frac{k^2}{m_{n-1}'^2}$$

$$\eta_{n-1} = \frac{1}{\tau_{n-1}} \left( \frac{c_n}{1-l} - \frac{k^2}{m_{n-1}'^2} \gamma_{n-2} - \frac{w_{n-2}^2}{m_{n-1}'^2} c_{n-1} \right)$$

$$s_{n-1}^2 = \frac{k^2 w_{n-2}^2 + w_{n-2}^2 \frac{\sigma^2}{(1-l)^2} + \frac{\sigma^2}{(1-l)^2} k^2}{m_{n-1}'^2 \tau_{n-1}^2}$$

(7)

$\int$ over $h_{n-1}$ : $\quad v^2 = x_{n-1}^2 + s_{n-1}^2$

$$o^2 = \frac{x_{n-1}^2 s_{n-1}^2 + s_{n-1}^2 m_{n-1}^2 + m_{n-1}^2 x_{n-1}^2}{v^2}$$

$$\mu = \frac{s_{n-1}^2 u_{n-2} + x_{n-1}^2 \eta_{n-1}}{v^2}$$

$$P_{n-1} = P_{n-2} \cdot N \left( u_{n-2} - \eta_{n-1}, v^2 \right) \cdot N \left( \frac{c_{n-2} - \gamma_{n-2}}{\alpha_{n-2}} - \mu, o^2 \right) \cdot \frac{1}{\alpha_{n-2}(1-l)\tau_{n-1}}$$

### Likelihood calculation for directed motion

Similarly to the confined motion formula, the track probability can be expressed using a recurrence formula that depends on the parameters of the directed motion model, the localization error $\sigma$, the diffusion length $d$, the initial velocity $v$ and the speed of the velocity change $q$.

- Initialization:

Step0 :

$$s_0^2 = \sigma^2 + d^2$$
$$x_0^2 = v^2 + q^2$$
$$g_0^2 = \frac{q^2 s_0^2 + s_0^2 * v^2 + v^2 * q^2}{x_0^2}$$
$$u_0^2 = \sigma^2 + g_0^2$$
$$\alpha_0 = v^2 / x_0^2$$
$$\gamma_0 = c_0$$
$$h_0^2 = v^2$$

Step1 :

$$k_1 = \frac{c_1 - \gamma_0}{\alpha_0}$$
$$h_1^2 = \frac{g_0^2 + \sigma^2}{\alpha_0^2}$$
$$a_1 = \frac{\sigma^2 \alpha_0}{u_0^2} + 1$$
$$b_0 = \frac{g_0^2 c_1 + \sigma^2 \gamma_0}{u_0^2}$$
$$s_1^2 = \frac{\sigma^2 g_0^2 + g_0^2 d^2 + d^2 \sigma^2}{u_0^2 a_i^2}$$
$$\beta_1 = \frac{x_0^2}{x_0^2 + q^2}$$
$$t_1^2 = \beta_1 q^2$$
$$x_1^2 = x_0^2 + q^2$$
$$\alpha_1 = \frac{a_1 \beta_1 h_0^2}{h_0^2 + t_1^2}$$
$$\gamma_1 = b_0 + \frac{a_1 t_1^2 k_0}{h_0^2 + t_1^2}$$
$$g_1^2 = a_i^2 \frac{t_1^2 s_1^2 + s_1^2 h_0^2 + h_0^2 t_1^2}{h_0^2 + t_1^2}$$
$$o_1 = \frac{k_1}{\beta_1}$$
$$q_1^2 = \frac{h_1^2 + t_1^2}{\beta_1^2}$$
$$P_1 = \frac{1}{\alpha_0 \beta_1}$$

(8)

- Recurrence:

$$u_i = \frac{c_i - \gamma_{i-1}}{\alpha_{i-1}}$$

$$h_i^2 = \frac{g_{i-1}^2 + \sigma^2}{\alpha_{i-1}^2}$$

$$a_i = \frac{\sigma^2 \alpha_{i-1}}{\sigma^2 + g_{i-1}^2} + 1$$

$$b_i = (g_{i-1}^2 c_i + \sigma^2 \gamma_{i-1})/(\sigma^2 + g_{i-1}^2)$$

$$s_i^2 = \frac{\sigma^2 g_{i-1}^2 + g_{i-1}^2 d^2 + d^2 \sigma^2}{a_i^2(\sigma^2 + g_{i-1}^2)}$$

$$k_i = \frac{h_i^2 o_{i-1} + q_{i-1}^2 u_i}{h_i^2 + q_{i-1}^2}$$

$$w_i^2 = \frac{h_i^2 q_{i-1}^2}{h_i^2 + q_{i-1}^2}$$

$$x_i^2 = x_{i-1}^2 + q^2$$

$$\beta_i = \frac{x_{i-1}^2}{x_i^2}$$

$$t_i^2 = \beta_i q^2$$

$$\alpha_i = \frac{a_i \beta_i w_i^2}{w_i^2 + t_i^2}$$

$$\gamma_i = b_i + \frac{a_i t_i^2 k i}{w_i^2 + t_i^2}$$

$$g_i^2 = a_i^2 \frac{t_i^2 s_i^2 + s_i^2 w_i^2 + w_i^2 t_i^2}{w_i^2 + t_i^2}$$

$$o_i = k_i/\beta_i$$

$$q_i^2 = \frac{w_i^2 + t_i^2}{\beta_i^2}$$

$$P_i = \frac{P_{i-1}}{\alpha_{i-1}\beta_i} N(u_i - o_{i-1}, h_i^2 + q_{i-1}^2) \tag{9}$$

- Final steps (steps $i = n - 2$ and $i = n - 1$):

$$u_{n-2} = \frac{c_{n-2} - \gamma_{n-3}}{\alpha_{n-3}}$$

$$h_{n-2}^2 = \frac{\sigma^2 + g_{n-3}^2}{\alpha_{n-2}^2}$$

$$a_{n-2} = \frac{\sigma^2 \alpha_{n-3}}{\sigma^2 + g_{n-3}^2} + 1$$

$$b_{n-2} = \frac{g_{n-3}^2 c_{n-2} + \sigma^2 \gamma_{n-3}}{\sigma^2 + g_{n-3}^2}$$

$$s_{n-2}^2 = \frac{\sigma^2 g_{n-3}^2 + g_{n-3}^2 d^2 + d^2 \sigma^2}{a_{n-2}^2(\sigma^2 + g_{n-3}^2)}$$

$$k_{n-2} = \frac{h_{n-2}^2 o_{n-2} + q_{n-3}^2 u_{n-2}}{h_{n-1}^2 + q_{n-3}^2}$$

$$w_{n-2}^2 = \frac{h_{n-2}^2 * q_{n-3}^2}{h_{n-2}^2 + q_{n-3}^2}$$

$$y_{i-1}^2 = w_{n-2}^2 + x_{n-3}^2$$

$$P_{n-2} = \frac{P_{n-3}}{\alpha_{n-2}} N(u_{n-2} - o_{n-3}, h_{n-2}^2 + q_{n-3}^2) N(k_{n-1}, w_{n-2}^2 + x_{n-3}^2)$$

$$\gamma_{n-2} = b_{n-2} + \frac{a_{n-2} x_{n-3}^2 k_{n-2}}{w_{n-2}^2 + x_{n-3}^2}$$

$$g_{n-2}^2 = a_{n-2}^2 \frac{x_{n-3}^2 s_{n-2}^2 + s_{n-2}^2 w_{n-2}^2 + w_{n-2}^2 x_{n-3}^2}{w_{n-2}^2 + x_{n-3}^2}$$

$$P_{n-1} = P_{n-2} N(c_{n-1} - \gamma_{n-1}, \sigma^2 + g_{n-1}^2) \tag{10}$$

