## [Editor Report · eLife Assessment]

In this **valuable** contribution, the authors present a novel and versatile probabilistic tool for classifying tracking behaviors and understanding parameters for different types of single-particle motion. The software package will be broadly applicable to single-particle tracking studies. The methodology has been **convincingly** tested by computational comparisons and experimental data, although the mathematical foundation for the hypothesis testing method can be further strengthened.

---

## [Referee Report · Reviewer #1 (Public review)]

Summary:

Weiss and co-authors presented a versatile probabilistic tool. aTrack helps in classifying tracking behaviors and understanding important parameters for different types of single particle motion types: Brwonian, Confined, or Directed motion. The tool can be used further to analyze populations of tracks and the number of motion states. This is a stand-alone software package, making it user-friendly for a broad group of researchers.

Strengths:

This manuscript presents a novel method for trajectory analysis.

Comments on revisions:

The authors have strengthened and improved the manuscript

---

## [Referee Report · Reviewer #2 (Public review)]

Summary:

The authors present a software package "aTrack" for identification of motion types and parameter estimation in single-particle tracking data. The software is based on maximum likelihood estimation of the time-series data given an assumed motion model and likelihood ratio tests for model selection. They characterized the performance of the software mostly on simulated data and showed that it is applicable to experimental data.

Strengths:

Although many tools exist in the single-particle tracking (SPT) field, this particular software package is developed using an innovative mathematical model and a probabilistic approach. It also provide inference of motion types, which are critical to answer biological questions in SPT experiments.

(1) The authors adopt a novel mathematical framework, which is unique in the SPT field.

(2) The authors have validated their method extensively using simulated tracks and compared to existing methods when appropriate.

(3) The code is freely available

Weaknesses:

The authors did a good job during the revision to address most of the weaknesses in my (as well as other reviewer's) first round of review. Nevertheless, the following issue is still not fully addressed.

The hypothesis testing method presented here lacks rigorous statistical foundation. The authors improved on this point after the revision, but in their newly added SI section "Statistical Test", only justified their choices using "hand-waving" arguments (i.e. there is not a single reference to proper statistical textbooks or earlier works in this important section). I understand that sometimes mathematical rigor comes later after some intuition-guided choices of critical parameters seems to work, but nevertheless need to point it out as a remaining weakness.

---

## [Author Response]

The following is the authors’ response to the original reviews.

**Reviewer #1 (Public Review):**
Summary:Weiss and co-authors presented a versatile probabilistic tool. aTrack helps in classifying tracking behaviors and understanding important parameters for different types of single particle motion types: Brownian, Confined, or Directed motion. The tool can be used further to analyze populations of tracks and the number of motion states. This is a stand-alone software package, making it user-friendly for a broad group of researchers.Strengths:This manuscript presents a novel method for trajectory analysis.Weaknesses:(1) In the results section, is there any reason to choose the specific range of track length for determining the type of motion? The starting value is fine, and would be short enough, but do the authors have anything to report about how much is too long for the model?

We chose to test the range of track lengths (five-to-hundreds of steps) to cover the broad range of scenarios arising from single proteins or fluorophores to brighter objects with more labels. While there is no upper-limit per se, the computation time of our method scales linearly with track length, 100 time-points takes ~2 minutes to run on a standard consumer-level desktop CPU. We have added the following sentence to note the time-cost with trajectory length:

“The recurrent formula enables our model computation time to scale linearly with the number of time points.”

(2) Robustness to model mismatches is a very important section that the authors have uplifted diligently. Understanding where and how the model is limited is important. For example, the authors mentioned the limitation of trajectory length, do the authors have any information on the trajectory length range at which this method works accurately? This would be of interest to readers who would like to apply this method to their own data.

We agree that limitations are important to estimate, and trajectory length is an important consideration when choosing how to analyze a dataset. We report the categorization certainty, i.e. the likelihood differences, for a range of track lengths (Fig. 2 a,c, Fig. 3c-d, and Fig. 4 c,g.).

For example, here are the key plots from Fig. 2 quantifying the relative likelihoods, where being within the light region is necessary. The light areas represent a useful likelihood ratio.

We only performed analysis up to track lengths of 600 time steps but parameter estimations and significance can only improve when increasing the track length as long as the model assumptions are verified. The broader limitations and future opportunities for new methods are now expanded upon in the discussion, for example switching between states and model and state and model ambiguities (bound vs very slow diffusion vs very slow motion).

(3) aTrack extracts certain parameters from the trajectories to determine the motion types. However, it is not very clear how certain parameters are calculated. For example, is the diffusion coefficient D calculated from fitting, and how is the confinement factor defined and estimated, with equations? This information will help the readers to understand the principles of this algorithm.

We apologize for the confusion. All the model parameters are fit using the maximum likelihood approach. To make this point clearer in the manuscript, we have made three changes:

(1) We modified the following sentence to replace “determined” with "fit”:

“Finally, Maximum Likelihood Estimation (MLE) is used to fit the underlying parameter value”

(2) We added the following sentence in the main text :

“In our model, the velocity is the characteristic parameter of directed motion and the confinement factor represents the force within a potential well. More precisely, the confinement factor $l$ is defined such that at each time step the particle position is updated by $l$ times the distance particle/potential well center (see the Methods section for more details).”.

(3) We have added a new section in the methods, called Fitting Method, where we have added the explanation below:

“For the pure Brownian model, the parameters are the diffusion coefficient and the localization error. For the confinement model, the parameters are the diffusion coefficient, the localization error, confinement factor, and the diffusion coefficientof the potential well. For the directed model, the parameters are the diffusion coefficient, the localization error, the initial velocity and the acceleration variance.

These parameters are estimated using the maximum likelihood approach which consists in finding the parameters that maximize the likelihood. We realize this fitting step using gradient descent via a TensorFlow model. All the estimates presented in this article are obtained from a single set of initial parameters to demonstrate that the convergence capacity of aTrack is robust to the initial parameter values.”

(4) The authors mentioned the scenario where a particle may experience several types of motion simultaneously. How do these motions simulated and what do they mean in terms of motion types? Are they mixed motion (a particle switches motion types in the same trajectory) or do they simply present features of several motion types? It is not intuitive to the readers that a particle can be diffusive (Brownian) and direct at the same time.

In the text, we present an example where one can observe this type of motion to help the reader understand when this type of motion can be met: “Sometimes, particles undergo diffusion and directed motion simultaneously, for example, particles diffusing in a flowing medium (Qian 1991).”

This is simulated by the addition of two terms affecting the hidden position variable before adding a localization term to create the observed variable. In the analysis, this manifests as non-zero values for the diffusion coefficient and the linear velocity. For example, Figure 4g and the associated text, where a single particle moves with a directed component and a Brownian diffusion component at each step.

We did not simulate transitions between types of motion. Switching is not treated by this current model; however, this limitation is described in the discussion and our team and others are currently working on addressing this challenge.

**Reviewer #2 (Public Review):**
Summary:The authors present a software package "aTrack" for identification of motion types and parameter estimation in single-particle tracking data. The software is based on maximum likelihood estimation of the time-series data given an assumed motion model and likelihood ratio tests for model selection. They characterized the performance of the software mostly on simulated data and showed that it is applicable to experimental data.Strengths:A potential advantage of the presented method is its wide applicability to different motion types.Weaknesses:(1) There has been a lot of similar work in this field. Even though the authors included many relevant citations in the introduction, it is still not clear what this work uniquely offers. Is it the first time that direct MLE of the time-series data was developed? Suggestions to improve would include (a) better wording in the introduction section, (b) comparing to other popular methods (based on MSD, step-size statistics (Spot-On, eLife 2018;7:e33125), for example) using the simulated dataset generated by the authors, (c) comparing to other methods using data set in challenges/competitions (Nat. Comm (2021) 12:6253).

We thank the reviewer for this suggestion and agree that the explanation of the innovative aspects of our method in the introduction was not clear enough. We have now modified the introduction to better explain what is improved here compared to previous approaches.

“The main innovations of this model are: (1) it uses analytical recurrence formulas to perform the integration step for complex motion, improving speed and accuracy; (2) it handles both confined and directed motion; (3) anomalous parameters, such as the center of the potential well and the velocity vector are allowed to change through time to better represent tracks with changing directed motion or confinement area; and lastly (4) for a given track or set of tracks, aTrack can determine whether tracks can be statistically categorized as confined or directed, and the parameters that best describe their behavior, for example, diffusion coefficient, radius of confinement, and speed of directed motion.”

Regarding alternatives, we compare our method in the text to the best-performing algorithm of the

2021 Anomalous Diffusion (AnDi) Challenge challenge mentioned by the reviewer in Figure 6 (RANDI, Argun et al, arXiv, 2021, Muñoz-Gil et al, Nat Com. 2021). Notably, both methods performed similarly on fBm, but ours was more robust in cases where there were small differences between the process underlying the data and the model assumptions, a likely scenario in real datasets. Regarding Spot-On, this was not mentioned as it only deals with multiple populations of Brownian diffusers, preventing a quantitative comparison.

(2) The Hypothesis testing method presented here has a number of issues: first, there is no definition of testing statistics. Usually, the testing statistics are defined given a specific (Type I and/or Type II) error rate. There is also no discussion of the specificity and sensitivity of the testing results (i.e. what's the probability of misidentification of a Brownian trajectory as directed? etc).

We now explain our statistical approach and how to perform hypothesis testing with our metric in a new supplementary section, Statistical test.

We use the likelihood ratio as a more conservative alternative to the p-value. In Fig S2, we show that our metric is an upper bound of the p-value and can be used to perform hypothesis testing with a chosen type I error rate.

Related, it is not clear what Figure 2e (and other similar plots) means, as the likelihood ratio is small throughout the parameter space. Also, for likelihood ratio tests, the authors need to discuss how model complexity affects the testing outcome (as more complex models tend to be more "likely" for the data) and also how the likelihood function is normalized (normalization is not an issue for MLE but critical for ratio tests).

We present the likelihood ratio as an upper bound of the p-value. Therefore, we can reject the null hypothesis if it is smaller than a given threshold, e.g. 0.05, but this number should be decreased if multiple tests are performed. The colorscale we show in the figure is meant to highlight the working range (light), and ambiguous range (dark) of the method.

As the reviewer mentions, we expect the alternative hypothesis to result in higher likelihoods than the simpler null hypothesis for null hypothesis tracks, but, as seen in the Fig S2, the likelihood ratio of a dataset corresponding to the null hypothesis is strongly skewed toward its upper limit 1. This means that for most of the tracks, the likelihood is not (or little) affected by the model complexity. The likelihoods of all the models are normalized so their integrals over the data equals 1/A with A the area of the field of view which is independent of the model complexity.

(3) Relating to the mathematical foundation (Figure 1b). The measured positions are drawn as direct arrows from the real position states: this infers instantaneous localization. In reality, there is motion blur which introduces a correlation of the measured locations. Motion blur is known to introduce bias in SPT analysis, how does it affect the method here?

The reviewer raises an important point as our model does not explicitly consider motion blur. We have now added a paragraph that presents how our model performs in case of motion blur in the section called Robustness to model mismatches. This section and the corresponding new Supplemental Fig. S7 demonstrate that the estimated diffusion length is accurate so long as the static localization error is higher than the dynamic localization error. If the dynamic localization error is higher, our model systematically underestimates the diffusion length by a factor 0.81 = (2/3)^0.5^ which can be corrected for with an added post-processing step.

(4) The authors did not go through the interpretation of the figure. This may be a matter of style, but I find the figures ambiguous to interpret at times.

We thank the reviewer for their feedback on improving the readability. To avoid overly repetitive and lengthy sections of text, we have opted for a concise approach. This allows us to present closely related panels at the same point in the text, while not ignoring important variations and tests. Considering this feedback and the reviewers, we have added more information and interpretation throughout our manuscript to improve interpretability.

(5) It is not clear to me how the classification of the 5 motion types was accomplished.

We have modified the specific text related to this figure to describe an illustrative example to show how one could use aTrack on a dataset where not that much is known: First, we present the method to determine the number of states; second, we verify the parameter estimates correspond to the different states.

Classifying individual tracks is possible. While not done in the section corresponding to Fig. 5, this is done in Fig. 7 and a new supplementary plot, Fig. S9b (shown below). In brief, this is accomplished with our method by computing the likelihood of each track given each state. The probability that a given track is in state k equals the likelihood of the track given the state divided by the sum of the likelihoods given the different states.

(6) Figure 3. Caption: what is ((d_{est}-0.1)/0.1)? Also panel labeled as "d" should be "e".

Thank you for bringing these errors to our attention, the panel and caption have been corrected.

**Reviewer #3 (Public Review):**
Summary:In this work, Simon et al present a new computational tool to assess non-Brownian single-particle dynamics (aTrack). The authors provide a solid groundwork to determine the motion type of single trajectories via an analytical integration of multiple hidden variables, specifically accounting for localization uncertainty, directed/confined motion parameters, and, very novel, allowing for the evolution of the directed/confined motion parameters over time. This last step is, to the best of my knowledge, conceptually new and could prove very useful for the field in the future. The authors then use this groundwork to determine the motion type and its corresponding parameter values via a series of likelihood tests. This accounts for obtaining the motion type which is statistically most likely to be occurring (with Brownian motion as null hypothesis). Throughout the manuscript, aTrack is rigorously tested, and the limits of the methods are fully explored and clearly visualised. The authors conclude with allowing the characterization of multiple states in a single experiment with good accuracy and explore this in various experimental settings. Overall, the method is fundamentally strong, wellcharacterised, and tested, and will be of general interest to the single-particle-tracking field.Strengths:(1) The use of likelihood ratios gives a strong statistical relevance to the methodology. There is a sharp decrease in likelihood ratio between e.g. confinement of 0.00 and 0.05 and velocity of 0.0 and 0.002 (figure 2c), which clearly shows the strength of the method - being able to determine 2nm/timepoint directed movement with 20 nm loc. error and 100 nm/timepoint diffusion is very impressive.

We apologize for the confusion, the directed tracks in Fig 2 have no Brownian-motion component, i.e. D=0. We have made this clearer in the main text. Specifically, this section of the text refers to a track in linear motion with 2 nm displacements per step. With 70 time points (69 steps), a single particle which moved from 138 nm with a localization error of 20 nm (95% uncertainty range of 80 nm) can be statistically distinguished from slow diffusive motion.

In Fig. 4g, we explore the capabilities of our method to detect if a diffusive particle also has a directed motion component.

(2) Allowing the hidden variables of confinement and directed motion to change during a trajectory (i.e. the q factor) is very interesting and allows for new interpretations of data. The quantifications of these variables are, to me, surprisingly accurate, but well-determined.(3) The software is well-documented, easy to install, and easy to use.Weaknesses:(1) The aTrack principle is limited to the motions incorporated by the authors, with, as far as I can see, no way to add new analytical non-Brownian motion. For instance, being able to add a dynamical stateswitching model (i.e. quick on/off switching between mobile and non-mobile, for instance, repeatable DNA binding of a protein), could be of interest. I don't believe this necessarily has to be incorporated by the authors, but it might be of interest to provide instructions on how to expand aTrack.

We agree that handling dynamic state switching is very useful and highlight this potential future direction in the discussion. The revised text reads:

“An important limitation of our approach is that it presumes that a given track follows a unique underlying model with fixed parameters. In biological systems, particles often transition from one motion type to another; for example, a diffusive particle can bind to a static substrate or molecular motor (46). In such cases, or in cases of significant mislinkings, our model is not suitable. However, this limitation can be alleviated by implicitly allowing state transitions with a hidden Markov Model (15) or alternatives such as change-point approaches (30, 47, 48), and spatial approaches (49).”

(2) The experimental data does not very convincingly show the usefulness of aTrack. The authors mention that SPBs are directed in mitosis and not in interphase. This can be quantified and studied by microscopy analysis of individual cells and confirming the aTrack direction model based on this, but this is not performed. Similarly, the size of a confinement spot in optical tweezers can be changed by changing the power of the optical tweezer, and this would far more strongly show the quantitative power of aTrack.

We agree with the reviewer and have revised the biological experiment section significantly to better illustrate the potential of aTrack in various use cases.

Now, we show an experiment to quantify the effect of LatA, an actin inhibitor, on the fraction of directed tracks obtained with aTrack. We find that LatA significantly decreases directed motion while a LatA-resistant mutant is not affected (Fig7a-c).

As suggested by the reviewer, we have expanded the optical tweezer experiment by varying the laser power. As expected, increasing the laser power decreases the confinement radius.

(3) The software has a very strict limit on the number of data points per trajectory, which is a user input. Shorter trajectories are discarded, while longer trajectories are cut off to the set length. It is not explained why this is necessary, and I feel it deletes a lot of useful data without clear benefit (in experimental conditions).

We thank the reviewer for this recommendation; we have now modified the architecture of our model to enable users to consider tracks of multiple lengths. Note that the computation time is proportional to the longest track length times the number of tracks.

**Reviewer #2 (Recommendations For The Authors):**
Develop a better mathematical foundation for the likelihood ratio tests.

We added more explanation of the likelihood ratio tests and their interpretation a new section entitled Statistical test in the supplementary information to address this recommendation.

Place this work in clearer contexts.

We have now revised the introduction to better contextualize this work.

Improve manuscript clarity.

Based on reviewer feedback and input from others, we have addressed this point throughout the article to improve readability.

Make the code available.

The code is available on https://github.com/FrancoisSimon/aTrack, now including code for track generation.

**Reviewer #3 (Recommendations For The Authors):**
(1) I believe the underlying model presented in Figure 1 is of substantial impact, especially when considering it as a simulation tool. I would suggest the authors make their method also available as a simulator (as far as I can tell, this is not explicitly done in their code repository, although logically the code required for the simulator should already be in the codebase somewhere).

Thank you for this suggestion, the simulation scripts are now on the Github repository together with the rest of the analysis method. https://github.com/FrancoisSimon/aTrack

(2) The authors should explore and/or discuss the effects of wrong trajectory linking to their method. Throughout the text, fully correct trajectory linking is assumed and assessed, while in real experiments, it is often the case that trajectory linking is wrong, e.g. due to blinking emitters, imaging artefacts, high-density localizations, etc etc. This would have a major impact on the accuracy of trajectories, and it is extremely relevant to explore how this is translated to the output of aTrack.

As the reviewer notes, our current model does not account for track mislinking. This limits the method to data with lower fluorophore-densities, which is the typical use-case for SPT. We have added a brief description of the issue into the discussion of limitations.

(3) aTrack only supports 2D-tracking, but I don't believe there is a conceptual reason not to have this expanded to three dimensions.

The stand-alone software is currently limited to 2D tracks, however, the aTrack Python package works for any number of dimensions (i.e. 1-3). Note that since the current implementation assumes a single localization error for all axes, more modifications may be required for some types of 3D tracking. See https://github.com/FrancoisSimon/aTrack for more details about aTrack implementations.

(4) Crucial information is missing in the experimental demonstrations. Especially in the NP-bacteria dataset, I miss scalebars, and information on the number of tracks. It is not explained why 5 different states are obtained - especially because I would naively expect three states: immobile NPs (e.g. stuck to glass), diffusing NPs, and NPs attached to bacteria, and thus directed. Figure 7e shows three diffusive states (why more than one?), no immobile states (why?), and two directed states (why?).

We thank the reviewer for pointing out these issues. We have now added scalebars and more experimental details to the figure and text as well as modifying the plot to more clearly emphasize the directed nanoparticles that are attached to cells from the diffusive nanoparticles.

Likely, our focal plane was too high to see the particles stuck on glass. The multiple diffusive states may be caused by different sizes of nanoparticle complexes, the multiple directed states can be caused by the fact that directed motion of the cell-attached-nanoparticles occasionally shows drastic changes of orientations. We have also clarified in the text how multiple states can help handle a heterogeneous population as was shown by Prindle et al. 2022, Microbiol Spectr. The characterization and phenotyping of microbial populations by nanoparticle tracking was published in Zapata et al. 2022, Nanoscale.

(5) I don't think I agree that 'robustness to model mismatches' is a good thing. Very crudely, the fact that aTrack finds fractional Brownian motion to be normal Brownian motion is technically a downside - and this should be especially carefully positioned if (in the future) a fractional Brownian motion model would be added to aTrack. I think that the author's point can be better tested by e.g. widely varying simulated vs fitted loc precision/diffusion coefficient (which are somewhat interchangeable).

In this context, our intention in describing the robustness to “model mismatches” refers to classifying subdiffusion as subdiffusive irrespective of the exact subdiffusion motion physics (as well as superdiffusion), that is, to use aTrack how MSD analysis is often deployed. This is important in the context of real-world applications where simple mathematical models cannot perfectly represent real tracks with greater complexity.

Inevitably, some fraction of tracks with a pure Brownian motion may appear to match with a fractional Brownian motion, and thus statistical tests are needed to determine if this is significant. In general, aTrack finds fBm to be normal Brownian motion only when the anomalous coefficient is near 1, i.e. when the two models are indeed the same. When analysing fBm tracks with anomalous coefficients of 0.5 or 1.5, aTrack find that these tracks are better explained by our confined diffusion model or directed motion model, respectively (Please see Fig. 6a, copied below).

To better clarify our objective, the section now has a brief introduction that reads:

“One of the most important features of a method is its robustness to deviations from its assumptions. Indeed, experimental tracking data will inevitably not match the model assumptions to some degree, and models need to be resilient to these small deviations.”

Smaller points:(1) It is not clear what a biological example is of rotational diffusion.

We modified the text to better explain the use of rotational diffusion.

(2) The text in the section on experimental data should be expanded and clarified, there currently are multiple 'floating sentences' that stop halfway, and it does not clearly describe the biological relevance and observed findings.

We thank the reviewer for pointing out this issue. We have reworked the experimental section to better and more clearly explain the biological relevance of the findings.

(3) Caption of figure 3: 'd' should be 'e'.(4) Caption of Figure 7: log-likelihood should be Lconfined - Lbrownian, I believe.(5) Equation number missing in SI first sentence.(6) Supplementary Figure 1 top part access should be Lc-Lb instead of Ld-Lb.

We have made these corrections, thank you for bringing them to our attention.